# Sensitivity of asymmetric Oxygen Minimum Zones to mixing intensity and stoichiometry in the tropical Pacific using a basin-scale model (OGCM-DMEC V1.4)

Kai Wang[1], Xiujun Wang[1,2*], Raghu Murtugudde[2], Dongxiao Zhang[3], Rong-Hua Zhang[4]

[1]College of Global Change and Earth System Science, Beijing Normal University, Beijing 100875, China
[2]Earth System Science Interdisciplinary Center, University of Maryland, College Park, Maryland 20740, USA
[3]JISAO, University of Washington and NOAA, Pacific Marine Environmental Laboratory, Seattle, Washington 98115, USA
[4]Institute of Oceanology, Chinese Academy of Sciences, Qingdao, Shandong 266071, China

*Correspondence to*: Xiujun Wang (xwang@bnu.edu.cn)

**Abstract.** The tropical Pacific Ocean holds the two largest Oxygen Minimum Zones (OMZs) in the world's oceans, showing a prominent hemispheric asymmetry, with a much stronger and broader OMZ north of the equator. However, many models have difficulties in reproducing the observed asymmetric OMZs in the tropical Pacific. Here, we apply a fully coupled basin-scale model to evaluate the impacts of stoichiometry and the intensity of vertical mixing on the dynamics of OMZs in the tropical Pacific. We first utilize observational data of dissolved oxygen (DO) to calibrate and validate the basin-scale model. Our model experiments demonstrate that an enhanced vertical mixing combined with a reduced O:C utilization ratio can significantly improve our model capability of reproducing the asymmetric OMZs. Our study shows that DO concentration is more sensitive to biological processes over 200-400 m but to physical processes below 400 m. Applying an enhanced vertical mixing causes a modest increase in physical supply (1-2 mmol $m^{-3}$ $yr^{-1}$) and a small increase (<0.5 mmol $m^{-3}$ $yr^{-1}$) in biological consumption over 200-1000 m whereas applying a reduced O:C utilization ratio leads to a large decrease (2-8 mmol $m^{-3}$ $yr^{-1}$) in both biological consumption and physical supply in the OMZs. Our analyses suggest that biological consumption (greater rate to the south than to the north) cannot explain the asymmetric distribution of mid-depth DO in the tropical Pacific, but physical supply (stronger vertical mixing to the south) play a major role in regulating the asymmetry of the tropical Pacific's OMZs. This study also highlights the important roles of physical and biological interactions and feedbacks in contributing to the asymmetry of OMZs in the tropical Pacific.

## 1 Introduction

Photosynthesis and respiration are important processes in all ecosystems on earth, with carbon and oxygen being the two main elements. The carbon cycle has garnered much attention, with significant advances in both observations (Feely et al., 1999; Takahashi et al., 2009) and modelling (DeVries et al., 2019; Le Quéré et al., 2010; McKinley et al., 2016) of biological processes (e.g., uptake of $CO_2$ and respiration) and physical/chemical processes (e.g., carbon fluxes between the

atmosphere, land and ocean). However, the oxygen cycle has received much less attention despite its large role in the earth system (Breitburg et al., 2018; Oschlies et al., 2018).

Dissolved oxygen (DO) is a sensitive indicator of physical and biogeochemical processes in the ocean thus a key parameter for understanding the ocean's role in the climate system (Stramma et al., 2010). In addition to photosynthesis and respiration,

the distribution of DO in the world's oceans is also regulated by air-sea gas exchange, ocean circulation and ventilation (Bettencourt et al., 2015; Bopp et al., 2002; Levin, 2018). Unlike most dissolved nutrients that display an increase in concentration with depth, DO concentration is generally low at mid-depth of the oceans. The most remarkable feature in the oceanic oxygen dynamics is the so-called Oxygen Minimum Zone (OMZ) that is often present below 200 m in the open oceans (Karstensen et al., 2008; Stramma et al., 2008). Previous studies have used the isoline of 20 mmol m$^{-3}$ as the

boundary of the OMZ for the estimation of OMZ volume (Bettencourt et al., 2015; Bianchi et al., 2012; Fuenzalida et al., 2009), and also as an up limit for the suboxic water (Wright et al., 2012).

The world's two largest OMZs are observed in the Eastern Tropical North Pacific (ETNP) and South Pacific (ETSP), showing a peculiar asymmetric structure across the equator, i.e., a much larger volume of suboxic water (<20 mmol m$^{-3}$) to

the north than to the south (Bettencourt et al., 2015; Paulmier and Ruiz-Pino, 2009). It is known that OMZs are caused by continuous biological consumption of DO associated with remineralization of organic matter (OM), and weak physical supply of DO due to sluggish subsurface ocean circulation and ventilation (Brandt et al., 2015; Czeschel et al., 2011; Kalvelage et al., 2015). Although there have been a number of observation-based analyses addressing the dynamics of OMZs in the tropical Pacific during the past decade (Czeschel et al., 2012; Garçon et al., 2019; Schmidtko et al., 2017; Stramma et

al., 2010), our understanding is limited on the underlying mechanisms that regulate the asymmetry of mid-depth DO (Oschlies et al., 2018; Stramma et al., 2012).

Large-scale physical-biogeochemical models have become a useful tool to investigate the responses of OMZs to climate change (Duteil and Oschlies, 2011; Ward et al., 2018; Williams et al., 2014). However, many models are unable to reproduce

the observed patterns of asymmetric OMZs in the tropical Pacific (Cabre et al., 2015; Shigemitsu et al., 2017), which may be due to "unresolved ocean transport processes, unaccounted for variations in respiratory oxygen demand, or missing biogeochemical feedbacks" (Oschlies et al., 2018). A common problem is that the two asymmetric OMZs merge into one in most models due to overestimated OMZ volume in the tropical Pacific, which may be related to the regulation of physical supply and/or biological demand (Cabre et al., 2015; Shigemitsu et al., 2017). While some studies suggest that a realistic

representation of circulation and ventilation with a high-resolution ocean model is critical to the simulation of asymmetric OMZs in the tropical Pacific (Berthet et al., 2019; Busecke et al., 2019), other modelling studies have demonstrated that physical processes (e.g., vertical mixing) play a major role in regulating the distribution of mid-depth DO (Brandt et al., 2015; Llanillo et al., 2018). On the other hand, there have been advances in understanding of biogeochemical regulation on

DO consumption, i.e., oxygen-restricted remineralization of organic matter (Kalvelage et al., 2015). Hence, it's necessary to
carry out integrative model-data studies to improve model capacity of simulating the dynamics of the tropical OMZs, and to better understand the relative roles of physical and biological processes in regulating the asymmetry of the tropical Pacific OMZs.

A basin-scale ocean general circulation model coupled with a dynamic marine ecosystem-carbon model (OGCM-DMEC) has been developed for the tropical Pacific, which has demonstrated capability of reproducing observed spatial and temporal variations of physical, nutrient and carbon fields in the upper ocean (Wang et al., 2008; Wang et al., 2015; Wang et al., 2009b), and the distribution of particulate organic carbon (POC) and export production below 200 m (Yu et al., 2021). In this study, we conduct model sensitivity experiments and evaluations to examine the responses of mid-depth DO and the sources/sinks to parameterizations of two key processes (i.e., oxygen-restricted remineralization and vertical mixing). We first carry out model calibration and validation using observational data of basin-scale DO concentration and oxygen consumption in the water column of the south tropical Pacific to improve the simulation of OMZs in the tropical Pacific. Then, we analyse the impacts of new parameterizations on biological consumption and physical supply and their relative contributions to the dynamics of mid-depth DO. The objective of this study is to advance our model capacity to simulate the oceanic oxygen cycle, and to identify the mechanisms driving the asymmetric OMZs in the tropical Pacific.

## 2. Model description

### 2.1 Ocean physical model

The basin-scale OGCM is a reduced-gravity, primitive-equation, sigma-coordinate model and it is coupled to an advective atmospheric model (Murtugudde et al., 1996). There are 20 layers with variable thicknesses and a total depth of ~1200 m in the OGCM. The mixed layer (the upper-most layer) depth is determined by the Chen mixing scheme (Chen et al., 1994), which varies from 10 m to 50 m on the equator. The remaining layers in the euphotic zone are approximately 10 m in thickness. The vertical resolution is approximately 30-50 m in the core OMZ (at ~300-500 m). The model domain is between 30ºS and 30ºN for the Pacific, with the western boundary closed (i.e., no representation of the Indonesian throughflow). Zonal resolution is 1º, and meridional resolution varies from 0.3º-0.6º over 15ºS-15ºN (1/3° over 10°S-10ºN) to 2º in the southern and northern "sponge layers" (the 25º-30º bands) where temperature, salinity, nutrients and DO are gradually relaxed back towards the observed climatological seasonal means. The boundary conditions of temperature, salinity, nitrate and DO are from the World Ocean Atlas, 2013 (WOA2013: http://www.nodc.noaa.gov/OC5/woa13/pubwoa13.html), and boundary condition for dissolved iron is based on limited field data, and is represented by a linear regression against temperature (see details in Christian et al., 2001). Such model configuration may have a disadvantage for longer simulations and analyses, but has the advantage in reproducing the spatial patterns of most physical and biogeochemical fields.

The model is forced by the surface momentum, heat and freshwater fluxes: climatological monthly means of solar radiation and cloudiness, and interannual 6-day means of precipitation and surface wind stress. Precipitation is from ftp://ftp.cdc.noaa.gov/Datasets/gpcp. Wind stresses are from the National Centers for Environmental Prediction (NCEP) reanalysis (ftp://ftp.cdc.noaa.gov/Datasets/ncep.reanalysis2.dailyavgs/gaussian_grid). Air temperature and humidity above the ocean surface are computed by the atmospheric mixed layer model. Initial conditions are obtained from the outputs of an interannual hindcast simulation over 1948-2000, which itself is initialized from a 30-year spin up with climatological forcing, followed by two 40-year interannual simulations. The initial conditions for the spin up are from the WOA2013, iron concentration for the spin up is initialized from limited field data collected in the tropical Pacific (Johnson et al., 1997). We carry out an interannual simulation for the period of 1978-2010, and analyse the mean states from model simulations over the period of 1991-2010.

## 2.2 Ocean biogeochemical model

The DMEC model consists of eleven components: small (S) and large (L) sizes of phytoplankton ($P_S$ and $P_L$), zooplankton ($Z_S$ and $Z_L$) and detritus ($D_S$ and $D_L$), dissolved organic nitrogen (DON), ammonium, nitrate, dissolved iron, and DO (Figure 1). Phytoplankton growth is co-limited by nitrogen and iron, which is critical in the tropical Pacific. The model simulates the iron cycle using variable Fe:N ratios, and incorporates atmospheric iron input. All biological components use nitrogen as their unit, in which sources/sinks are determined by biological and chemical processes in addition to the physical processes (circulation and vertical mixing) that are computed by the OGCM.

In this model, net community production (NCP) is computed as:

$$NCP = 6.625(\mu_S P_S + \mu_L P_L - r_S Z_s - r_L Z_L - c_{DON} DON - c_{DS} D_S - c_{DL} D_L) \tag{1}$$

where 6.625 is the C:N ratio, $\mu$ the constant of phytoplankton growth, $r$ the constant of zooplankton respiration, $c$ the constant of detritus decomposition or DON remineralization. The equations for biogeochemical processes and model parameters are given in Appendix A and B. There are changes in some parameters comparing with those used in Wang et al. (2008), which are based on our further model calibration and validation for chlorophyll (Wang et al., 2009a), nitrogen cycle (Wang et al., 2009b) and carbon cycle (Wang et al., 2015).

Recently, we have made further improvements in the parameterizations of detritus decomposition and DON remineralization (eq. B21-B23), which result from the first round of model calibration on DO distribution using WOA2013. In brief, the constant $c_{DON}$ decreases exponentially with depth over 100-1000 m in this study. The differences in the related parameters are given in Appendix C.

## 2.3 Computation of oxygen sources and sinks

The time evolution of DO is regulated by physical, biological and chemical processes:

$$\frac{\partial O_2}{\partial t} = -u\frac{\partial O_2}{\partial x} - v\frac{\partial O_2}{\partial y} - w\frac{\partial O_2}{\partial z} + O_{mix} - O_{bio} + O_{gas} \tag{2}$$

where $u$, $v$, and $w$ are zonal, meridional, and vertical velocity, respectively.

The term $O_{gas}$, flux of $O_2$ from the atmosphere into the surface ocean, is computed as:

$$O_{gas} = (O_{Sat} - O_2)K_0 \tag{3}$$

where $O_{sat}$ is the $O_2$ saturation, a function of temperature and salinity (Weiss, 1970), and $K_0$ the gas transfer velocity that is a function of wind speed ($u_s$) and SST according to Wanninkhof (1992):

$$K_0 = 0.31u_s^2\sqrt{\frac{S_c}{S_{c20}}} \tag{4}$$

where $S_c$ and $S_{c20}$ are the Schmidt number at SST and $20^{\circ}$ C, respectively:

$$S_c = 1953 - 128T + 3.99T^2 - 0.05T^3 \tag{5}$$

The biological source/sink term $O_{bio}$ is computed as follows:

$$O_{bio} = R_{OC}\text{NCP} \tag{6}$$

where $R_{OC}$ is the O:C utilization ratio (set to 1.3 in reference simulation, according to the Redfield ratio). Below the euphotic zone, biological consumption ($O_{cons}$) is determined by detritus decomposition and DON remineralization:

$$O_{cons} = 6.625(c_{DON}DON + c_{DS}D_S + c_{DL}D_L)R_{OC} \tag{7}$$

in which DON remineralization is dominant because DON pool is several times greater than detritus (Wang et al., 2008).

The vertical mixing term $O_{mix}$ is calculated by three subroutines. Briefly, the first one computes convection to remove instabilities in the water column, and the second one determines the mixed layer depth based on the available surface turbulent kinetic energy. The third one computes partial vertical mixing ($K_z$) between two adjacent layers to relieve gradient Richardson ($R_i$) number instability, which is calculated as follows:

$$K_z = \left(1 - \left(\frac{Ri}{0.7}\right)^\lambda\right)(Ri < 0.7) \tag{8}$$

$$K_z = 0 \ (Ri \geq 0.7) \tag{9}$$

where the mixing parameter $\lambda$ is set to 1. Partial vertical mixing is the dominant process influencing physical supply of DO in the intermediate waters.

Rate of total physical supply ($O_{sup}$) below the euphotic zone is computed as:

$$O_{sup} = \frac{\Delta O}{\Delta t} + O_{cons} \tag{10}$$

Total physical supply consists of meridional, zonal and vertical advection, and vertical mixing. The advection terms are computed from the corresponding velocity and DO gradient, and the vertical mixing term is calculated as the residue.

## 3. Model experiments

### 3.1 Evaluation of DO distribution from the reference run

We first evaluate simulated DO for the tropical Pacific Ocean using the outputs from the OGCM-DMEC V1.4 (hereafter reference run), which use the same set of parameters as Yu et al. (2021). We focus on model-data comparisons over 200-400 m, 400-700 m and 700-1000 m, that broadly represent the upper OMZ, core OMZ and lower OMZ, respectively. The WOA2013 data shows a much larger area of suboxic waters (<20 mmol m$^{-3}$) in the ETNP than in the ETSP over 200-400 m and 400-700 m (Figure 2a and 2c), but no suboxic water over 700-1000 m (Figure 2e). Although the reference run produces two OMZs off the equator over 200-400 m (Figure 2b), the sizes of suboxic water are much larger in the reference run than those in the WOA2013 data. The reference run significantly over-estimates the size of suboxic water and underestimates DO concentration over 400-700 m (Figure 2d). The difference between WOA2013 and the reference run is small over 700-1000 m, except in the eastern tropical Pacific (Figure 2f).

### 3.2 Sensitivity experiments

Given that the DO concentration at mid-depth is regulated by physical supply and biological consumption, the underestimation of mid-depth DO would be a result of overestimation of consumption associated with DON remineralization and/or underestimation of supply. The reference run applies a value of zero for background diffusion (see eq. 9). However, a previous modelling study has demonstrated that vertical background diffusion is an important process for DO supply at mid-depth (Duteil and Oschlies, 2011). Accordingly, we conduct a sensitivity experiment to test a set of values for background diffusion (Kb as 0.1, 0.25 and 0.5 cm$^2$ s$^{-1}$). The addition of background diffusion is only applied to the two key variables (DO and DON) in this analysis to eliminate any potential interactions and feedbacks between various physical and biogeochemical processes (note: our model experiments show no significant effects on modelled DO dynamics with the background diffusion applied to the nutrients).

There have been several advances in understanding of oxygen consumption. For example, recent studies have shown that the O:C utilization ratio varies greatly across different basins, e.g., from 0.6 to 2.1 in the Pacific (Moreno et al., 2020; Tanioka and Matsumoto, 2020), and rates of DOM remineralization or oxygen consumption are influenced by oxygen level, i.e., a reduction under low DO conditions (Beman et al., 2020; Bertagnolli and Stewart, 2018; Sun et al., 2021). Using the field data collected from mid-depth (~350 m) in the Peruvian OMZ (Kalvelage et al., 2015),we derive a kinetics function between oxygen consumption rate and DO concentration, which yields two values for the half saturation constant ($K_m$), i.e., 6.9 and 18.7 mmol m$^{-3}$ (Figure 3). By applying this function, O:C utilization ratio in equation 7 becomes variable and is also

reduced (i.e., $R_{oc} = 1.3 \frac{DO}{DO+Km}$), with lower ratios in low-DO waters. Accordingly, our sensitivity experiments include a few simulations with a reduced O:C utilization ratio ($K_m$ as 6.9 or 18.7 mmol m$^{-3}$) and/or added background diffusion (Table S1).

Figure 4 illustrates that there is a larger volume of suboxic water located north of ~5°N and a smaller volume of suboxic water over 12°S-4°S in the WOA2013, which are separated by relatively higher DO (>20 mmol m$^{-3}$) water along the equator. There is an improvement in simulated DO with a reduced O:C utilization ratio (Figure 4b and 4c) and an enhanced vertical mixing (Figure 4d and 4g). Clearly, the combination of a reduced O:C utilization ratio and an enhanced vertical mixing leads to a further improvement in simulated mid-depth DO (Figure 4e, 4f, 4h and 4i). In particular, the combination of a stronger background diffusion with a smaller O:C utilization ratio (i.e., the Km18.7Kb0.5 run) results in the best simulation that reproduces the observed spatial distribution of mid-depth DO, especially the hemispheric asymmetry (i.e., a larger volume of suboxic water to the north but a smaller size of suboxic water to the south).

## 3.3 Model validation

To further demonstrate the impact of parameter choices, a few statistical measures are applied over 200-400 m, 400-700 m and 700-1000 m in the ETNP (165°W-90°W, 5°-20°N) and ETSP (110°W-80°W, 10°S-3°S). As shown in Table 1, bias and root mean square error (RMSE) are reduced in all sensitivity simulations, with the smallest values from Km18.7Kb0.5 run except over 700-1000 m in the ETNP. For example, both bias and RMSE in the Km18.7Kb0.5 run are smallest over 200-700 m in the ETNP (<7.8 and 10.2 mmol m$^{-3}$). Many current models show much larger RMSE (~20-80 mmol m$^{-3}$) from their simulations from mixed layer to 1000 m (Bao and Li, 2016; Cabre et al., 2015). Figure 5 also illustrates that the Km18.7Kb0.5 run produces the best outputs, according to the combined assessments of the correlation coefficient and normalized standard deviation (NSD), i.e., the distance to the observation. The distance is shortest over 400-700 m and 700-1000 m in both the ETNP and ETSP in the Km18.7Kb0.5 simulation. Clearly, the correlation coefficient was largest (0.38-0.99) in all sections; and the NSD is closest to 1 in the core OMZ of ETNP.

We also compare the sizes of suboxic and hypoxic waters between the model simulations and WOA2013 (Table 2). Sizes of suboxic and hypoxic waters are 5.97x10$^{15}$ m$^3$ and 19.98x10$^{15}$ m$^3$ in the north, and 1.43×10$^{15}$ m$^3$ and 7.12x10$^{15}$ m$^3$ in the south, respectively, in the WOA2013. While applying a reduced O:C utilization ratio and an enhanced vertical mixing can lead to an improvement in simulated OMZ volume, a significant improvement is obtained with the combination of a reduced O:C utilization ratio and an enhanced vertical mixing. Overall, the Km18.7Kb0.5 simulation has the best performance for reproducing the OMZ volumes, showing similar volumes for the suboxic water (5.55x10$^{15}$ m$^3$ to the north and 1.12x10$^{15}$ m$^3$ to the south) and the hypoxic water (20.91x10$^{15}$ m$^3$ and 7.39x10$^{15}$ m$^3$).

We then further validate the modelled DO from the best run (Km18.7Kb0.5), using the time series of the observed DO data (https://cchdo.ucsd.edu/). Figure 6 illustrates that the model can generally reproduce the vertical-zonal distributions of DO along 10°N and 17°S, spanning 1989 to 2009, particularly in the eastern tropical Pacific. For example, cruise data from the P04 line (during April-May, 1989) show a large area of low DO water from ~200 m to ~800 m (Figure 6a), and our model also predicts low DO water over ~200-700 m (Figure 6b).

## 4 Model results and discussions

In this section, we further compare the improved simulations (Km18.7, Kb0.5 and Km18.7Kb0.5) with the reference run to diagnose the responses of mid-depth DO, and biological consumption and physical supply to the improved parameterizations. We then analyse the interactions of physical and biogeochemical processes, and their impacts on the mid-depth DO. Finally, we explore the underlying mechanisms regulating the asymmetry of OMZs in the tropical Pacific.

### 4.1 Changes of mid-depth DO due to a reduced O:C utilization ratio and an enhanced vertical mixing

We first compare the changes in DO concentration between the three simulations over 200-400 m, 400-700 m, and 700-1000 m (Figure 7). Clearly, applying a reduced O:C utilization ratio causes an increase of DO in all three layers, with the greatest increase (>6 mmol m$^{-3}$) in the 200-400 m layer (Figure 7a), followed by a modest increase (~3-6 mmol m$^{-3}$) over 400-700 m (Figure 7d). Although the increase is generally smaller in the 700-1000 m layer (Figure 7g) than in the 400-700 m layer

(Figure 7d), the increase is greater in the north OMZ over 700-1000 m than over 400-700 m. Applying an enhanced vertical mixing results in a small increase of DO (~2-5 mmol m$^{-3}$) in the 10°S-10°N band over 200-400 m (Figure 7b), but a large increase (~5-15 mmol m$^{-3}$) in majority of the basin over 400-700 m and 700-1000 m (Figure 7e and 7h).

Overall, mid-depth DO shows an increase with the combination of a reduced O:C utilization ratio and an enhanced vertical
mixing (Figure 7c, 7f & 7i). A great increase of DO (>15 mmol m$^{-3}$) occurs over most of the basin over 400-700 m, mainly in the central tropical Pacific over 200-400 m, but in a few small areas over 700-1000 m. The spatial pattern and magnitude of DO increase resulting from the combination of a reduced O:C utilization ratio and an enhanced vertical mixing, have a large similarity to those with a reduced O:C utilization ratio for the 200-400 m layer (Figure 7a), but to those under an enhanced vertical mixing below 400 m (Figure 7e & 7h). For example, the relative increase of DO is similarly larger in the
north OMZ over 200-400 m under a reduced O:C utilization ratio with and without the addition of background diffusion, and over 700-1000 m under an enhanced vertical mixing (i.e., with additional background diffusion) with and without the change in the O:C utilization ratio. Our analyses suggest that the dominant process regulating the DO dynamics is biological consumption over 200-400 m, but physical supply over 400-1000 m.

**4.2 Effects of a reduced O:C utilization ratio and an enhanced vertical mixing on consumption and supply**

To better understand the effects of changes in the biological and/or physical parameters on the DO dynamics, we evaluate the responses of biological consumption and physical supply. As illustrated in Figure 8, changes in biological consumption caused by a reduced O:C utilization ratio are almost identical with or without background diffusion. In particular, biological consumption shows a large decrease (~2-8 mmol m$^{-3}$ yr$^{-1}$) over 200-400 m (Figure 8b), and a relatively small decrease (~0.2-1.0 mmol m$^{-3}$ yr$^{-1}$) over 400-700 m, with the largest decrease in the north OMZ (Figure 8e). There is a very small change in

biological consumption over 700-1000 m, i.e., a decrease of <0.1 mmol m$^{-3}$ yr$^{-1}$ over much of the basin but an increase of <0.1 mmol m$^{-3}$ yr$^{-1}$ in some parts of subtropical region (Figure 8h). On the other hand, applying an enhanced vertical mixing leads to a small increase (<0.2 mmol m$^{-3}$ yr$^{-1}$) in biological consumption in all three layers, with a relatively larger increase in the north OMZ (Figure 8c, 8f and 8i).

Figure 9 shows the effects of a reduced O:C utilization ratio and an enhanced vertical mixing on physical supply. With the combination of a reduced O:C utilization ratio and an enhanced vertical mixing, physical supply shows a small increase (by ~0.2-1.0 mmol m$^{-3}$ yr$^{-1}$) in the whole basin over 700-1000 m (Figure 9g) and only outside the OMZs over 400-700 m (Figure 9d), but a relatively larger decrease (by ~0.2-6 mmol m$^{-3}$ yr$^{-1}$ ) in the OMZs over 200-700 m (Figure 9a and 9d). Clearly, applying an enhanced vertical mixing leads to an increase of physical supply over majority of the basin, with a greater

increase over 400-1000 m (~0.2-1.0 mmol m$^{-3}$ yr$^{-1}$) than over 200-400 m (~0-0.4 mmol m$^{-3}$ yr$^{-1}$) (Figure 9c, 9f and 9i). However, applying a reduced O:C utilization ratio causes a large decrease in physical supply above 700 m, with a greater decrease over 400-700 m in the OMZs (~0.2-6 mmol m$^{-3}$ yr$^{-1}$), and very small changes (<0.2 mmol m$^{-3}$ yr$^{-1}$) over 700-1000 m (Figure 9b, 9e and 9h). Overall, the rate of physical supply is determined largely by vertical mixing over 700-1000 m, by both vertical mixing and biological consumption over 400-700 m, but by consumption over 200-400 m, implying complex

physical-biological interactions and feedbacks in the tropical Pacific OMZs.

**4.3 Interactive effects of physical and biological processes on the source and sink of mid-depth DO**

Our further analyses show an increase in physical supply under an enhanced vertical mixing in most parts of the 200-1000 m layer, with larger values below the OMZs particularly to the south (Figure 10a). Applying an enhanced vertical mixing also results in a generally small increase in biological consumption (Figure 10b). The small increase in consumption is

attributable to the increase in DON concentration (Figure 10c) that results from the enhancement in vertical mixing. Clearly, there is an overall increase in net flux, with the largest increases occurring mainly outside the OMZs below ~400 m (Figure 10d).

As expected, applying a reduced O:C utilization ratio results in a decrease in consumption in the suboxic waters (Figure 10f),

with a greater decrease in the north OMZ than in the south OMZ. Interestingly, physical supply shows an overall decrease in

the water column under a reduced O:C utilization ratio, with a greater decrease in the upper OMZs (Figure 10e). A decreased rate of consumption leads to a large increase in DON concentration, with a greater increase in the north OMZ than in the south OMZ (Figure 10g). Net flux shows a small increase in the whole basin under a reduced O:C utilization ratio, with a greater increase over ~200-400 m (Figure 10h).


The combination of an enhanced vertical mixing and a reduced O:C utilization ratio results in an increase of supply below the OMZs but a decrease of supply inside of OMZs (Figure 10i). There is an overall decrease of biological consumption in the water column, with a greater decrease in the upper OMZs (Figure 10j), which is similar to the changes under a reduced O:C utilization ratio (Figure 10f). DON concertation shows a greater increase in the north OMZ than in the south OMZ under the combination of a reduced O:C utilization ratio and an enhanced vertical mixing (Figure 10k), which is similar to the changes in DON under a reduced O:C utilization ratio (Figure 10g). Applying a reduced O:C utilization ratio combined with an enhanced vertical mixing leads to an increase in net flux over 200-1000 m, with a larger increase outside of OMZs (Figure 10l), which is much greater than that under a reduced O:C utilization ratio (Figure 10h), and also greater than that with an enhanced vertical mixing particularly in the lower part of OMZs (Figure 10d).


There is evidence that physical and biogeochemical processes have multiple interactions with impacts on various physical, chemical and biological fields which in turn have implications for the DO dynamics (Breitburg et al., 2018; Duteil and Oschlies, 2011; Oschlies et al., 2018). For example, observational and modelling studies show that changes in vertical mixing intensity can affect the distribution of DOM thus oxygen consumption at mid-depth (Duteil and Oschlies, 2011; Talley et al., 2016). On the other hand, applying a smaller O:C utilization ratio leads to lower consumption rates (Moreno et al., 2020), thus to a relatively higher DO concentration in the OMZs. Therefore, changes in the consumption caused by an enhanced vertical mixing and/or a reduced O:C utilization ratio would alter the gradients of DO concentration in the water column thus change the intensity of vertical mixing inside and around the OMZs.

Our analyses also show that both changes in supply and consumption with improved parameterizations of both vertical mixing and remineralization of DOM (i.e., Km18.7Kb0.5) are quite different from the sums of changes caused by single parameter change, particularly in the OMZs (Figure 10m and 10n), indicating strong physical and biological interactions with positive feedbacks in the low-DO waters. Clearly, the interactions have a relatively larger effect on physical supply because of its sensitivity to changes in both physical and biological parameters. As a result, the interactive effects result in 310 an overall increase in net flux in the OMZs (Figure 10p).

Physical supply can be divided into horizontal advection, vertical advection, and vertical mixing. Figure S1 shows that vertical mixing is the dominant process for DO supply, particularly above ~600 m in the OMZs. Other modelling studies have also demonstrated the dominant role of vertical mixing in supplying oxygen from the thermocline to OMZs (Duteil et

al., 2020; Llanillo et al., 2018). Our model simulation with the combination of an enhanced vertical mixing and a reduced O:C utilization ratio compares well with other modeling results (Duteil et al., 2020; Shigemitsu et al., 2017), in simulating meridional and zonal advections, and vertical mixing for DO transport (see Figure S2), which gives us confidence for the evaluation of different responses of key physical components to an enhanced vertical mixing and a reduced O:C utilization ratio.

As shown in Figure 11, there is no clear pattern in the response of advective transport, with very small values ($< \sim 1$ mmol m$^{-3}$ yr$^{-1}$) over the entire basin. However, DO supply by vertical mixing shows a strong response to different model parameterizations, with similar patterns as those in total supply (see Figure 10). Including background diffusion leads to a large increase of vertical mixing ($>1$ mmol m$^{-3}$ yr$^{-1}$) over most of the basin, with a greater increase mainly below the OMZs (Figure 11c). On the other hand, applying a reduced O:C utilization ratio causes a small increase in vertical mixing of DO ($<0.5$ mmol m$^{-3}$ yr$^{-1}$) outside of suboxic waters but a large decrease ($\sim 2$-8 mmol m$^{-3}$ yr$^{-1}$) inside of the suboxic waters (Figure 11g). A significant decrease in vertical mixing ($>3$ mmol m$^{-3}$ yr$^{-1}$) is mainly found above $\sim 400$ m in the OMZs, which corresponds to the decrease in vertical gradient of DO concentration (Figure 11h).

Vertical mixing of DO shows an increase ($\sim 1$-2 mmol m$^{-3}$ yr$^{-1}$) outside of the OMZs and a decrease ($\sim 2$-8 mmol m$^{-3}$ yr$^{-1}$) inside of the OMZs in response to the combination of an enhanced background diffusion and a reduced O:C utilization ratio (Figure 11k), which is similar to the net response of vertical mixing to the changes caused by individual parameters (see Figure 11c and 11g). However, the combined effects exceed the sum of two individual responses in the south OMZ and the lower part of the north OMZ (Figure 11o). An early study has demonstrated that applying an enhanced background diffusion can lead to an increase not only in vertical mixing of DO directly, but also in biological consumption caused by enhanced export production in the tropical Pacific OMZs (Duteil and Oschlies, 2011), which in turn changes the vertical gradient of DO concentration, thus affects the intensity of vertical mixing.

### 4.4 Impacts of biological and physical processes on asymmetric OMZs

There is evidence of asymmetric features in many biogeochemical fields in the tropical Pacific. For example, POC flux at 500 m is greater in the northern tropical Pacific ($\sim 4$ mmol C m$^{-2}$ d$^{-1}$) (Van Mooy et al., 2002) than in the southern tropical Pacific ($<1$ mmol C m$^{-2}$ d$^{-1}$) (Pavia et al., 2019). Similarly, our regional model reproduces an asymmetric pattern for POC flux, with larger values to the north than to the south. Field studies have reported an asymmetry in DOM distribution over $\sim 200$-1000 m in the central-eastern tropical Pacific, i.e., higher levels of DON and DOC to the north than to the south (Hansell, 2013; Libby and Wheeler, 1997; Raimbault et al., 1999). Our best model simulation (i.e., the Km18.7Kb0.5 simulation) also reveals an asymmetric DON below 300 m, i.e., $\sim 6$-8 mmol m$^{-3}$ in the ETNP and $\sim 3$-5 mmol m$^{-3}$ in the ETSP (Figure S3a). However, an earlier field study reported higher rates of organic carbon remineralization over 200-1000 m to

the south (~2-10 mmol m$^{-3}$ yr$^{-1}$ ) than to the north (~1-6 mmol m$^{-3}$ yr$^{-1}$) in the eastern/central tropical Pacific (Feely et al., 2004). Similarly, our model simulation also shows such an asymmetric feature in biological consumption over 300-600 m, i.e., ~2-4 mmol m$^{-3}$ yr$^{-1}$ in the ETSP and <1 mmol m$^{-3}$ yr$^{-1}$ in the ETNP (Figure S3b).

It appears that the asymmetric distributions differ largely between biological fields in the tropical Pacific. In particular, there are almost opposite patterns between oxygen consumption (or DOM remineralization) and DOM concentration, which may be attributed to the difference in rate of DOM remineralization between the north and south. Rate of DOM remineralization is determined not only by microbial activity, but also by the stoichiometry associated with microbial respiration (Wang et al., 2008; Zakem and Levine, 2019). Recent studies on the respiration quotient demonstrate that the O:C utilization ratio is lower to the north than to the south in the tropical Pacific (Tanioka and Matsumoto, 2020; Wang et al., 2019), which primarily reflects the difference in oxygen limitation on microbial respiration (Kalvelage et al., 2015). Apparently, the asymmetry in biological consumption (lower rate in the north than in the south) cannot explain the asymmetry in the tropical Pacific OMZs (i.e., lower DO levels to the north than to the south), indicating that other processes are responsible for the asymmetry.

Numerous studies have indicated that physical mixing is the only source of DO for the tropical OMZs (Brandt et al., 2015; Czeschel et al., 2012; Duteil et al., 2020). There is evidence that turbulent diffusion accounts for 89% of the net DO supply for the core OMZ of the southern tropical Pacific (Llanillo et al., 2018). Our model simulations show that zonal, meridional and vertical advections for DO supply are relatively weak (<2 mmol m$^{-3}$ yr$^{-1}$). However, the intensity of vertical mixing is much stronger (~2-6 mmol m$^{-3}$ yr$^{-1}$) at mid-depth, indicating that vertical mixing plays a bigger role in supplying DO into the OMZs.

Our further analyses show that the intensity of vertical mixing over 200-700 m is stronger to the south than to the north of the equator (Figure S2), which is consistent with some other modelling studies that report stronger DO supply via vertical mixing in the south OMZ than in the north OMZ in the tropical Pacific (Duteil, 2019; Shigemitsu et al., 2017). There is evidence that larger-scale mass transport due to circulation and ventilation is more efficient in the South Pacific than in the North Pacific (Kuntz and Schrag, 2018), and the transit time from the surface to the OMZ is much longer in the ETNP than in the ETSP (Fu et al., 2018). All these analyses indicate that vertical mixing is largely responsible for asymmetric distribution of mid-depth DO, and physical processes play a major role in shaping the asymmetry of the OMZs in the tropical Pacific.

### 4.5 Implications and limitations of the current research

There are inter-dependencies between physical and biogeochemical processes at mid-depth (Duteil and Oschlies, 2011; Gnanadesikan et al., 2012; Niemeyer et al., 2019), which can have influences on the asymmetry of OMZs in the tropical

Pacific. Our study shows that the rate of physical supply is sensitive to changes in vertical mixing below 400 m and biological consumption over 200-400 m. Since the contribution of physical supply to mid-depth DO flux exceeds that of biological consumption in the tropical Pacific (Llanillo et al., 2018; Montes et al., 2014), and the physical processes are more dominant in the ETSP, one may expect that physical-biological feedbacks are stronger to the south, which can lead to a relatively larger net flux into the south OMZ.

Physical and biogeochemical interactions are complex and region-specific, which produce direct and indirect effects on the sources and sinks of DO (Levin, 2018; Oschlies et al., 2018). Our study demonstrates that there is a much greater increase in net DO flux in the core OMZ to the south than to the north that results from these interactions and feedbacks (Fig. 10p). On the one hand, supply of DO is greater under stronger physical transport in the south tropical Pacific. On the other hand, stronger physical processes can also lead to higher levels of nutrients and biological production and thus enhanced export production and oxygen consumption at mid-depth (Duteil and Oschlies, 2011), which can offset the rate of physical supply. In addition, stronger physical processes can also result in strengthened transport of DO and OM out to other regions (Gnanadesikan et al., 2012; Yu et al., 2021), which have complex impacts on DO balance in the south OMZ.

To date, most regional to global models have difficulty in reproducing the observed asymmetric OMZs in the tropical Pacific (Cabre et al., 2015; Duteil, 2019), which is probably caused by misrepresentations of physical processes such as background diffusion at mid-depth and ocean circulation (Cabre et al., 2015; Duteil and Oschlies, 2011). In addition, model configuration such as vertical and horizontal resolutions can also influence physical transportations of DO (Busecke et al., 2019; Duteil et al., 2014), and distributions of nutrients (with impacts on biological production and DO consumption)(Berthet et al., 2019). Other possible causes may be associated with the ocean-atmosphere interactions and feedbacks due to the uncertainties in atmospheric forcing fields (Duteil, 2019; Ridder and England, 2014; Stramma et al., 2012).

Although there have been advances in our understanding of the regulation of DO consumption by biogeochemical processes, large scale models do not have representative processes due to various reasons. For example, there is evidence of DO depletion at mid-depth caused by zooplankton migration (Bianchi et al., 2013; Kiko et al., 2017), and there are strong interactions and feedbacks between carbon, nitrogen and oxygen cycles in marine ecosystems. Limited studies indicate that O:C:N utilization ratios during microbial respiration vary largely in the water column (Moreno et al., 2020; Zakem and Levine, 2019). Nitrogen cycling (e.g., oxidation, nitrification and denitrification) not only has impacts on oxygen consumption/production but is also influenced by the oxygen level (Beman et al., 2021; Kalvelage et al., 2013; Oschlies et al., 2019; Sun et al., 2021). However, little attention has been paid to understanding the coupling of carbon and oxygen cycles; the available data are also not sufficient for the parameterizations of relevant processes, which has hampered our ability to assess the impacts of those biogeochemical processes on oxygen fields. Future observational and modelling studies are needed not only to improve our knowledge on the coupling of carbon, nitrogen and oxygen cycles in the ocean, but also

to advance our understanding on the physical and biogeochemical interactions and feedbacks associated with marine stoichiometry.

## 5. Conclusion

In this paper, we use a basin scale model to investigate the impacts of parameterizations of vertical mixing and DOM remineralization on the dynamics of mid-depth DO, and analyse the underlying mechanisms for asymmetric OMZs in the
tropical Pacific. Our results show that the model is capable of reproducing the observed DO distributions and asymmetric OMZs with the combination of an enhanced vertical mixing and a reduced O:C utilization ratio that causes an increase in DO concentration (or net flux) at mid-depth. Overall, applying an enhanced vertical mixing makes a larger contribution to the increase of DO below ~400 m, and the contribution from a reduced O:C utilization ratio is greater over 200-400 m.

Our analyses demonstrate that there is a modest increase in physical supply and a small increase in biological consumption with an enhanced vertical mixing, and the increase in consumption is a result of the redistribution of DOM in the water column. On the other hand, applying a reduced O:C utilization ratio results in a large decrease in both biological consumption and physical supply in the OMZs (due to the changes in vertical DO gradients). These findings point to strong physical-biological interactions and feedbacks at mid-depth in the tropical Pacific.

This study suggests that biological consumption (i.e., greater rate to the south) cannot explain the asymmetric feature in the tropical Pacific OMZs (i.e., lower DO levels to the north), but physical processes (i.e., stronger vertical mixing to the south) play a major role in shaping the asymmetric OMZs of the tropical Pacific. In addition, the interactions between physical and biological processes are also stronger in the south OMZ than in the north OMZ, likely because physical supply is sensitive to
changes in the parameterizations of both vertical mixing and DOM remineralization. Further studies with improved approaches will enable a better understanding of the interactions and feedbacks between physical and biogeochemical processes.

## Appendix A: Model biogeochemical equations

### Phytoplankton equations

$$\frac{\partial P_S}{\partial t} = \mu_S P_S - g_{P_S}(1 - e^{-\Lambda P_S})Z_S - m_S P_S \tag{B1}$$

$$\frac{\partial P_L}{\partial t} = \mu_L P_L - g_{P_{L1}}(1 - e^{-\Lambda P_L})Z_L - g_{P_{L2}}(1 - e^{-\Lambda P_L})Z_S - m_L P_L \tag{B2}$$

### Zooplankton equations

$$\frac{\partial Z_S}{\partial t} = [\lambda(g_{P_S}(1 - e^{-\Lambda P_S}) + g_{P_{L2}}(1 - e^{-\Lambda P_L})) + g_{D_S}(1 - e^{-\Lambda D_S}) + g_{D_{L2}}(1 - e^{-\Lambda D_L}) - (r_s + \delta_s)]Z_S - g_{Z_S}(1 - e^{-\Lambda Z_S})Z_L \tag{B3}$$

$$\frac{\partial Z_L}{\partial t} = [\lambda(g_{P_{L1}}(1 - e^{-\Lambda P_L}) + g_{Z_S}(1 - e^{-\Lambda Z_S})) + g_{D_{L1}}(1 - e^{-\Lambda D_L}) - (r_L + \delta_L)]Z_L \tag{B4}$$

### Detritus equations

$$\frac{\partial D_S}{\partial t} = (m_S P_S + m_L P_L + (r_s Z_S + r_L Z_L)\chi)(1 - \gamma) - g_{D_S}(1 - e^{-\Lambda D_S})Z_S - (c_{D_S} + \omega_{D_S}h^{-1})D_S \tag{B5}$$

$$\frac{\partial D_L}{\partial t} = (1 - \lambda)[\left(g_{P_S}(1 - e^{-\Lambda P_S}) + g_{P_{L2}}(1 - e^{-\Lambda P_L})\right)Z_S + \left(g_{P_{L1}}(1 - e^{-\Lambda P_L}) + g_{Z_S}(1 - e^{-\Lambda Z_S})\right)Z_L] + \delta_S Z_S + \delta_L Z_L -$$
$$(c_{D_L} + \omega_{DL}h^{-1})D_L - g_{D_{L2}}(1 - e^{-\Lambda D_L})Z_S - g_{D_{L1}}(1 - e^{-\Lambda D_L})Z_L \tag{B6}$$

### DON equations

$$\frac{\partial DON}{\partial t} = (m_S P_S + m_L P_L + (r_s Z_S + r_L Z_L)\chi)\gamma + (c_{D_S}D_S + c_{D_L}D_L)\zeta - c_{DON}DON \tag{B7}$$

### Nutrients equations

$$\frac{\partial NO_3}{\partial t} = -\mu_S P_S \frac{N_{S\_UP}}{N_{S\_UP} + A_{UP}} - \mu_L P_L \frac{N_{L\_UP}}{N_{L\_UP} + A_{UP}} + \varphi NH_4 \tag{B8}$$

$$\frac{\partial NH_4}{\partial t} = -\mu_S P_S \frac{A_{UP}}{N_{S\_UP} + A_{UP}} - \mu_L P_L \frac{A_{UP}}{N_{L\_UP} + A_{UP}} + (r_s Z_S + r_L Z_L)(1 - \chi) + c_{DON}DON + (c_{D_S}D_S + c_{D_L}D_L)(1 - \zeta) - \varphi NH_4 \tag{B9}$$

$$\frac{\partial Fe}{\partial t} = -(\mu_S P_S R_S + \mu_L P_L R_L - s_{Fe}D_L Fe) + R_S[(r_s Z_S + r_L Z_L)(1 - \chi) + c_{DON}DON + c_{D_S}D_S + c_{D_L}D_L(1 - \zeta)] \tag{B10}$$

### Nitrogen uptake

$$N_{S\_UP} = \frac{NO_3}{K_{S\_NO_3} + NO_3}(1 - \frac{NH_4}{K_{NH_4} + NH_4}) \tag{B11}$$

$$N_{L\_UP} = \frac{NO_3}{K_{L\_NO_3} + NO_3}(1 - \frac{NH_4}{K_{NH_4} + NH_4}) \tag{B12}$$

$$A_{UP} = \frac{NH_4}{K_{NH_4} + NH_4} \tag{B13}$$

## Other equations

**Phytoplankton growth rate**

$$\mu_S = \mu_{S0}e^{k_T T}f(I)\psi_S(N,Fe) \tag{B14}$$

$$\mu_L = \mu_{L0}e^{k_T T}f(I)\psi_L(N,Fe) \tag{B15}$$

**Nutrient limitation**

$$\psi_S(N,Fe) = \min\left(\frac{NO_3+NH_4}{K_{S\_N}+NO_3+NH_4}, \frac{Fe}{K_{S\_Fe}+Fe}\right) \tag{B16}$$

$$\psi_L(N,Fe) = \min\left(\frac{NO_3+NH_4}{K_{L\_N}+NO_3+NH_4}, \frac{Fe}{K_{L\_Fe}+Fe}\right) \tag{B17}$$

**Light limitation**

$$f(I) = 1 - e^{-\frac{\alpha I}{\eta P_{MAX}}} \tag{B18}$$

**Light attenuation**

$$I(z) = I_0 exp^{-k_A Z} \tag{B19}$$

$$k_A = k_W + k_C \text{Chl} + k_D(D_S+D_L) \tag{B20}$$

**Detritus decomposition and DON remineralization**

$$c_{DS} = c_{DS0}e^{k_B(T-T_0)} \tag{B21}$$

$$c_{DL} = c_{DL0}e^{k_B(T-T_0)} \tag{B22}$$

$$c_{DON} = c_{DON0}e^{k_B(T-T_0)} \tag{B23}$$

**Phytoplankton carbon to chlorophyll ratio (η)**

$$\text{Chl} = \left(\frac{P_S}{\eta_S}+\frac{P_L}{\eta_L}\right)R_{C:N} \tag{B24}$$

$$\eta_S = \eta_{S0} - (\eta_{S0}-\eta_{MIN})\frac{lnI_0-lnI}{4.605} \tag{B25}$$

$$\eta_L = \eta_{L0} - (\eta_{L0}-\eta_{MIN})\frac{lnI_0-lnI}{4.605} \tag{B26}$$

$$\eta_{S0} = \eta_{S\_MAX} - k_{PS}\mu_S^* \tag{B27}$$

$$\eta_{L0} = \eta_{L\_MAX} - k_{PL}\mu_L^* \tag{B28}$$

$$\mu_S^* = \mu_{S0}e^{k_T T}\min\left(\frac{NO_3}{K_{S\_N}+NO_3}, \frac{Fe}{K_{S\_Fe}+Fe}\right) \tag{B29}$$

$$\mu_L^* = \mu_{L0} e^{k_T T} \min \left( \frac{NO_3}{K_{L\_N} + NO_3}, \frac{Fe}{K_{L\_Fe} + Fe} \right)$$ (B30)

500

## Appendix B: Model biogeochemical parameters

| Symbol | Parameter | Unit | Value |
|--------|-----------|------|-------|
| $m_S$ | Small phytoplankton mortality rate | d$^{-1}$ | 0.15 |
| $m_L$ | Large phytoplankton mortality rate | d$^{-1}$ | 0.35 |
| $r_S$ | Small zooplankton excretion rate | d$^{-1}$ | 0.53 |
| $r_L$ | Large zooplankton excretion rate | d$^{-1}$ | 0.44 |
| $\delta_S$ | Small zooplankton mortality rate | d$^{-1}$ | 0.12 |
| $\delta_L$ | Large zooplankton mortality rate | d$^{-1}$ | 0.12 |
| $g_{PS}$ | Maximum grazing rate for small phytoplanktion | d$^{-1}$ | 2.6 |
| $g_{PL1}$ | Maximum grazing rate for large phytoplanktion | d$^{-1}$ | 1.2 |
| $g_{ZS}$ | Maximum grazing rate for small zootoplanktion | d$^{-1}$ | 1.7 |
| $g_{PL2}$ | Maximum grazing rate for large phytoplanktion | d$^{-1}$ | 0.9 |
| $g_{DS}$ | Maximum grazing rate for small detritus | d$^{-1}$ | 1.0 |
| $g_{DL1}$ | Maximum grazing rate for large detritus | d$^{-1}$ | 3.0 |
| $g_{DL2}$ | Maximum grazing rate for large detritus | d$^{-1}$ | 1.5 |
| $\Lambda$ | Ivlev coefficient | (mmol m$^{-3}$)$^{-1}$ | 0.5 |
| $\lambda$ | Zooplankton assimilation coefficient | % | 75 |
| $\chi$ | Excretion coefficient | % | 55 |
| $\gamma$ | Dissolution coefficient | % | 90 |
| $\xi$ | Dissolution coefficient | % | 90 |
| $R_{C:N}$ | C:N ratio | mol:mol | 6.625 |
| $R_S$ | Fe:N ratio for small phytoplankton | $\mu$mol:mol | 15 |
| $R_L$ | Fe:N ratio for large phytoplankton | $\mu$mol:mol | 40 |
| $\eta_{S\_MIN}$ | Minimum PhyC:Chl ratio in small phytoplanktion | g:g | 30 |
| $\eta_{L\_MIN}$ | Minimum PhyC:Chl ratio in large phytoplanktion | g:g | 15 |
| $\eta_{S\_MAX}$ | Maximum PhyC:Chl ratio in small phytoplanktion | g:g | 200 |
| $\eta_{L\_MAX}$ | Maximum PhyC:Chl ratio in large phytoplanktion | g:g | 120 |
| $k_{PS}$ | Photoacclimation coefficient for small phytoplanktion | (g:g)d | 95 |
| $k_{PL}$ | Photoacclimation coefficient for large phytoplanktion | (g:g)d | 70 |
| $w_{DS}$ | Sinking velocity for small detritus | m d$^{-1}$ | 1 |

| | | | |
|---|---|---|---|
| $w_{DL}$ | Sinking velocity for large detritus | m d$^{-1}$ | 3.5 |
| $\varphi$ | Nitrification rate (when I<5 $\mu$molEm$^{-2}$ s$^{-1}$) | d$^{-1}$ | 0.04 |
| $s_{Fe}$ | Iron scavenge coefficient | d$^{-1}$ (nmol Fe m$^{-3}$)$^{-1}$ | 0.00001 |
| $\mu_{S0}$ | Maximum growth rate at 0°C for small phytoplankton | d$^{-1}$ | 0.58 |
| $\mu_{L0}$ | Maximum growth rate at 0°C for large phytoplankton | d$^{-1}$ | 1.16 |
| $k_T$ | Temp. Dependent coefficient for $\mu$ | °C$^{-1}$ | 0.06 |
| $K_{S\_N}$ | Half saturation constant for N limitation | mmol m$^{-3}$ | 0.3 |
| $K_{L\_N}$ | Half saturation constant for N limitation | mmol m$^{-3}$ | 0.9 |
| $K_{S\_Fe}$ | Half saturation constant for iron limitation | mmol m$^{-3}$ | 14 |
| $K_{L\_Fe}$ | Half saturation constant for iron limitation | mmol m$^{-3}$ | 150 |
| $K_{S\_NO3}$ | Half saturation constant for nitrate uptake | mmol m$^{-3}$ | 0.3 |
| $K_{L\_NO3}$ | Half saturation constant for nitrate uptake | mmol m$^{-3}$ | 0.9 |
| $K_{NH4}$ | Half saturation constant for ammonium uptake | mmol m$^{-3}$ | 0.05 |
| $\alpha$ | Initial slope of the P – I curve | mg C mg chl$^{-1}$($\mu mol$ E $m^{-2}$ $s^{-1}$)$^{-1}$ | 0.02 |
| $P_{MAX}$ | Maximum carbon specific growth rate | h$^{-1}$ | 0.036 |
| $k_W$ | Light attenuation constant for water | m$^{-1}$ | 0.028 |
| $k_C$ | Light attenuation constant for chlorophyll | m$^{-1}$ (mg chl m$^{-3}$)$^{-1}$ | 0.058 |
| $k_D$ | Light attenuation constant for detritus | m$^{-1}$ (mg chl m$^{-3}$)$^{-1}$ | 0.008 |
| $c_{DS0}$ | Small detritus decomposition rate at 0°C | d$^{-1}$ | 0.001 |
| $c_{DL0}$ | Large detritus decomposition rate at 0°C | d$^{-1}$ | 0.008 |

## Appendix C: Comparisons in biogeochemical parameters

| Symbol | Parameter | Unit | Yu et al. (2021) | This study |
|--------|-----------|------|------------------|------------|
| $T_0$ | Limit temperature | °C | 10 | 0 |
| $k_B$ | Temperature dependent coefficient | - | 0.002 | 0.001 |
| $C_{DON0}$ | DON remineralization constant | $d^{-1}$ | | |
| | 0-100 m | | 0.001 | 0.00075 |
| | 100-600 m | | 0.0002-0.001 | 0.00013-0.00075* |
| | 600-1000 m | | 0.0002 | 0.00003-0.00013* |

* $C_{DON0}$ decreases with depth by an exponential function.

*Code and data availability.* The exact version of the software code used to produce the results presented in this paper is archived on Zenodo (https://doi.org/10.5281/zenodo.5148146, Wang et al., 2021). Other code and data are available upon request from the authors. Request for materials should be addressed to X.J.W. (xwang@bnu.edu.cn).

*Author contributions.* X.J.W. and K.W. designed the study, performed the simulations and prepared the manuscript. R.M., D.X.Z. and R.H.Z. contributed to analysis, interpretation of results and writing.

*Competing interests.* The authors declare that they have no conflict of interest.

*Acknowledgements.* This work was supported by the Chinese Academy of Sciences' Strategic Priority Project (XDA1101010504). The authors wish to acknowledge the use of the Ferret (http://ferret.pmel.noaa.gov/Ferret/).

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

# Tables

**Table 1.** Bias and root mean square error (RMSE) for DO (mmol m$^{-3}$) comparisons between WOA2013 and model simulations averaged over 1991-2010 in the Eastern Tropical North Pacific (ETNP) and Eastern Tropical South Pacific (ETSP).

| Layers | Statistics | Ref | Kb0.1 | Kb0.25 | Kb0.5 | Km18.7 | Km6.9 | Km18.7 | Km6.9 Kb0.25 | Km6.9 Kb0.5 | Km18.7 Kb0.25 | Km18.7 Kb0.5 |
|---|---|---|---|---|---|---|---|---|---|---|---|---|
| ETNP (165°W-90°W, 5°N-20°N) | | | | | | | | | | | | |
| 200-400 m | Bias | -17.44 | -16.99 | -16.34 | -14.87 | -11.32 | -14.84 | -11.32 | -13.51 | -11.85 | -9.71 | -7.8 |
| | RMSE | 16.35 | 16.21 | 15.73 | 14.91 | 12.43 | 14.63 | 12.43 | 13.83 | 12.84 | 11.4 | 10.2 |
| 400-700 m | Bias | -16.35 | -14.37 | -11.85 | -7.5 | -12.51 | -14.95 | -12.51 | -9.98 | -5.39 | -6.88 | -2.04 |
| | RMSE | 10.6 | 9.54 | 8.26 | 6.73 | 8.45 | 9.83 | 8.45 | 7.49 | 6.38 | 6.5 | 6.78 |
| 700-1000 m | Bias | -9.22 | -6.61 | -3.58 | 0.62 | -5.99 | -8.32 | -5.99 | -2.71 | 1.38 | -5.75 | 3.27 |
| | RMSE | 5.1 | 3.06 | 2.93 | 6.52 | 2.64 | 4.29 | 2.64 | 3.59 | 7.19 | 5.39 | 9.08 |
| ETSP (110°W-80°W, 10°S-3°S) | | | | | | | | | | | | |
| 200-400 m | Bias | -7.09 | -6.84 | -6.43 | -5.39 | 0.19 | -3.91 | 0.19 | -2.84 | -1.13 | 2.09 | 4.85 |
| | RMSE | 7.39 | 7.20 | 6.83 | 5.98 | 2.36 | 4.46 | 2.36 | 3.69 | 2.86 | 3.27 | 5.51 |
| 400-700 m | Bias | -11.3 | -8.83 | -5.94 | -0.88 | -7.94 | -10.43 | -7.94 | -4.51 | 1.34 | -1.21 | 5.23 |
| | RMSE | 12.98 | 10.79 | 8.52 | 6.03 | 10.06 | 12.15 | 10.06 | 7.41 | 5.65 | 5.81 | 7.38 |
| 700-1000 m | Bias | -7.3 | -4.18 | -0.97 | 3.38 | -5.13 | -7.08 | -5.13 | -0.62 | 3.94 | 1.05 | 5.46 |
| | RMSE | 12.82 | 10.67 | 8.98 | 8.63 | 11.22 | 12.49 | 11.22 | 8.76 | 8.68 | 8.59 | 9.34 |

**Table 2.** Volumes (10$^{15}$ m$^3$) of suboxic and hypoxic waters from WOA2013 and model simulations.

| Regions | Waters | WOA2013 | Reference | Kb0.1 | Kb0.25 | Kb0.5 | Km6.9 | Km18.7 | Km6.9 Kb0.25 | Km6.9 Kb0.5 | Km18.7 Kb0.25 | Km18.7 Kb0.5 |
|---|---|---|---|---|---|---|---|---|---|---|---|---|
| North Pacific | Suboxic | 5.97 | 10.61 | 9.74 | 8.73 | 7.33 | 9.98 | 8.83 | 8.08 | 6.68 | 6.88 | 5.55 |
| | Hypoxic | 19.98 | 22.67 | 22.58 | 22.32 | 21.61 | 22.5 | 22.17 | 22.11 | 21.35 | 21.71 | 20.91 |
| South Pacific | Suboxic | 1.43 | 3.78 | 3.34 | 2.86 | 2.15 | 3.39 | 2.78 | 2.42 | 1.71 | 1.81 | 1.12 |
| | Hypoxic | 7.12 | 10.42 | 9.85 | 9.19 | 8.17 | 10.21 | 9.8 | 8.94 | 7.88 | 8.49 | 7.39 |

Suboxic: DO <20 mmol m$^{-3}$; Hypoxic: DO <60 mmol m$^{-3}$.

**Figures**

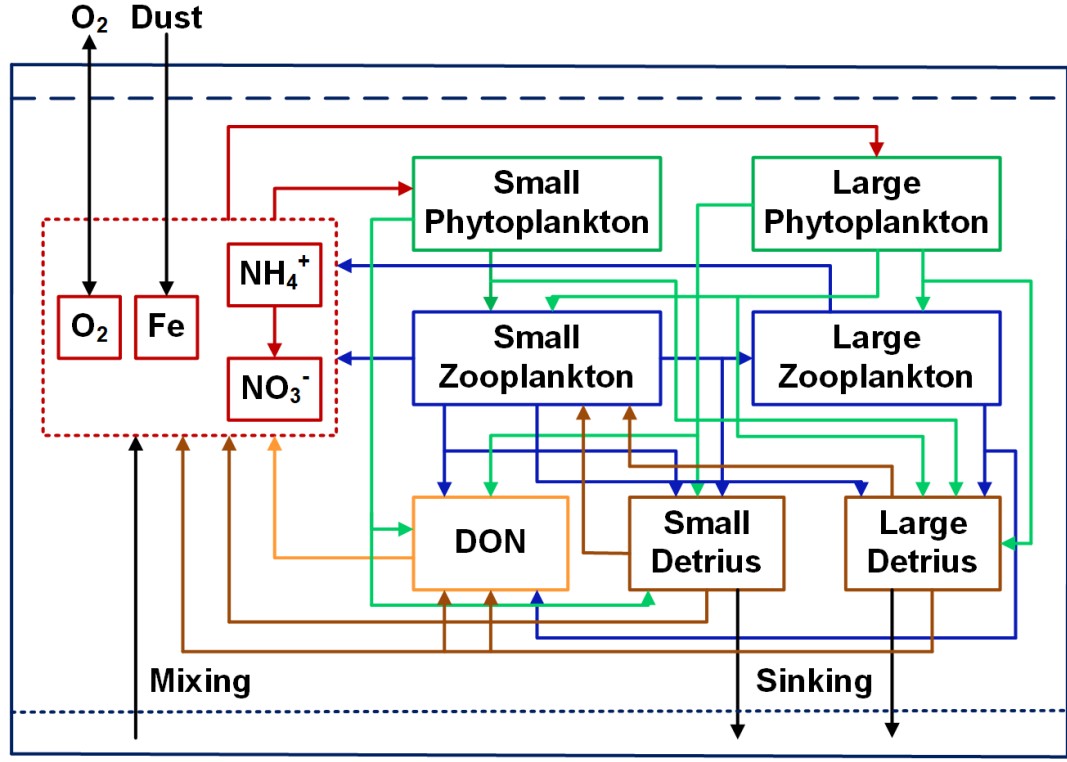


**Figure 1.** Flow diagram of the ecosystem model. Red, green, blue, yellow and brown lines and arrows denote fluxes originating from inorganic forms, phytoplankton, zooplankton, DON and detritus, respectively.

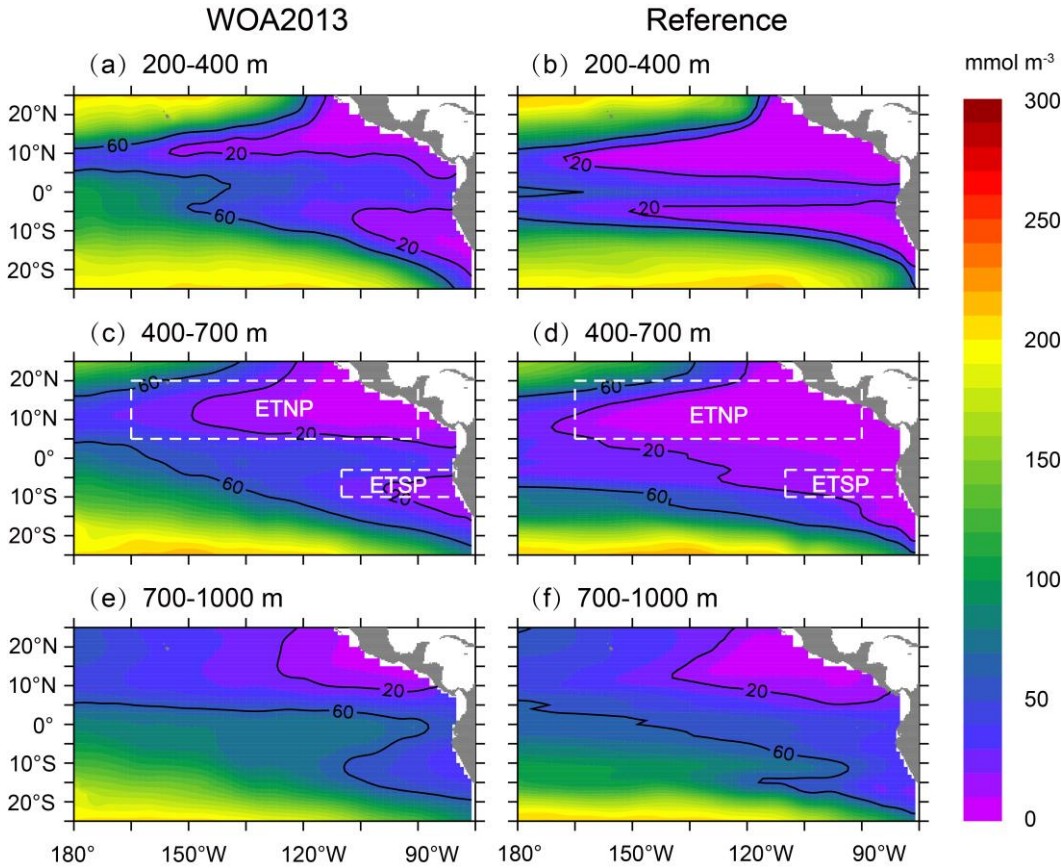

**Figure 2.** Comparisons of DO concentration between WOA2013 (left panel) and the reference run (right panel) over 1991-2010. White dash lines in **(c)** and **(d)** denote two boxes for ETNP (165°W-90°W, 5°N-20°N) and ETSP (110°W-80°W, 10°S-3°S).

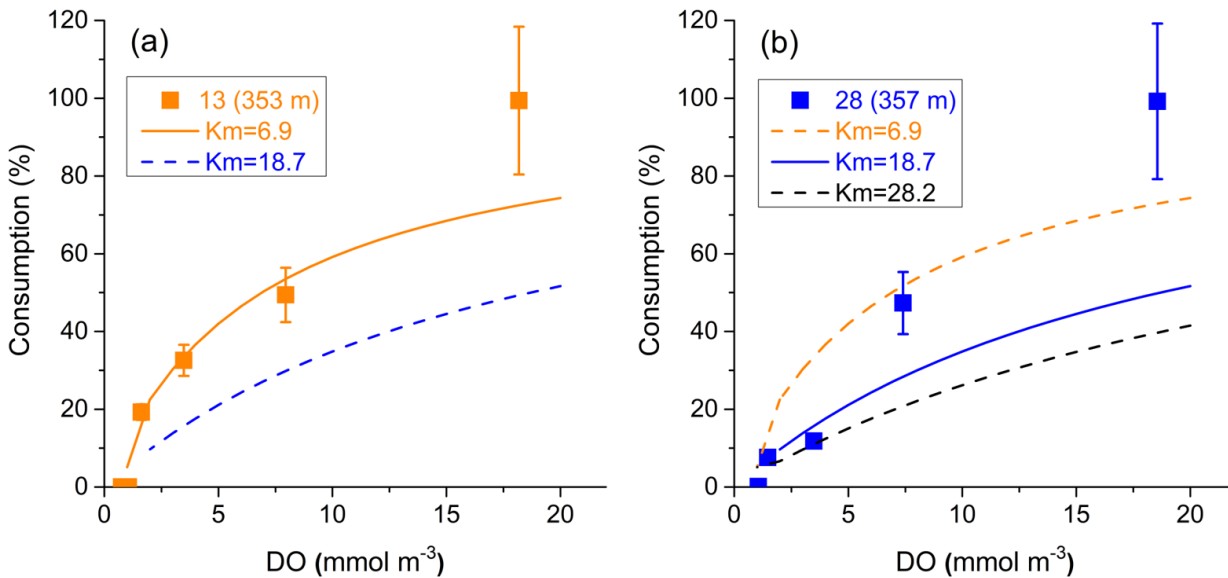

**Figure 3.** Relationship between biological consumption and DO concentration at **(a)** station 13 (353 m) and **(b)** station 28 (357 m) in the Peruvian OMZ. Km is the half saturation constant from the fitting curve derived from data of Kalvelage et al. (2015). For station 28, two Km values are derived using four (excluding the smallest value, blue curves) and five (black curve) data points, respectively. Since the curve with Km=28.2 is too far away from the most data points, the value 18.7 is selected as the half saturation constant.

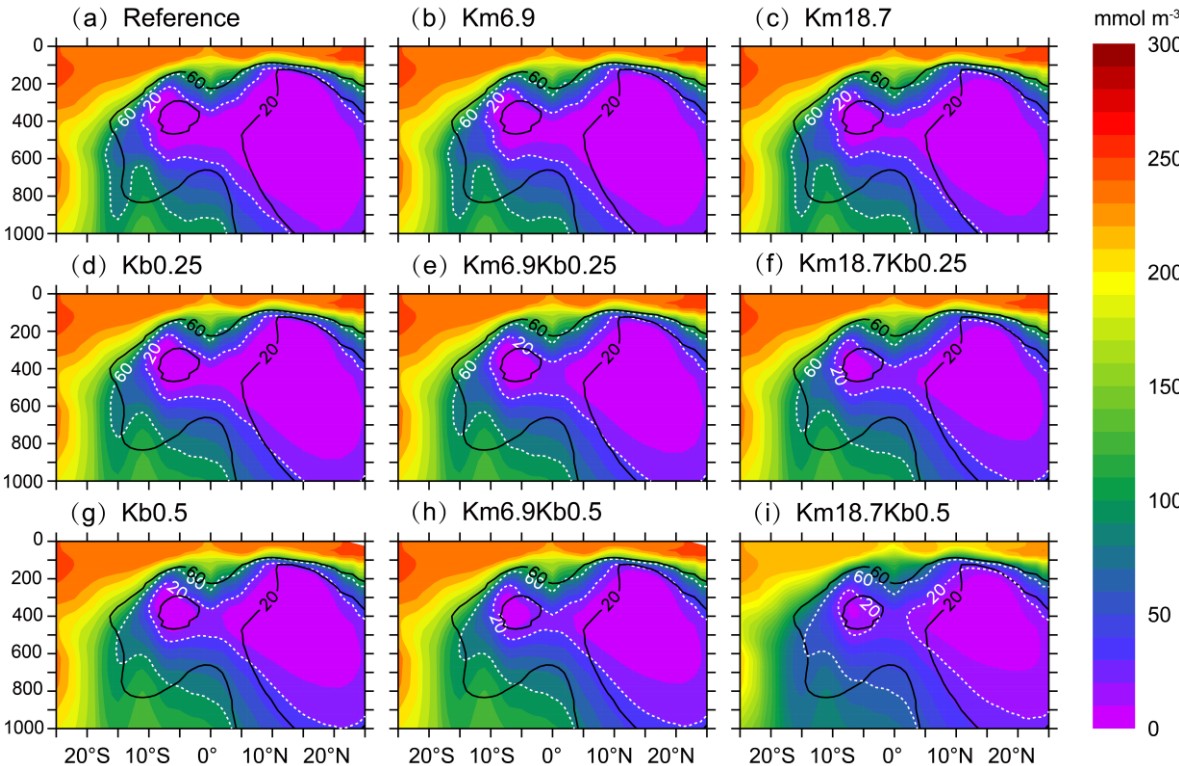


**Figure 4.** Vertical distribution of DO concentration over 120°W-90°W from different model simulations: **(a)** the reference run, **(b and c)** with a reduced O:C utilization ratio, **(d and g)** with an enhanced vertical mixing, and **(e, f, h, and i)** the combination of a reduced O:C utilization ratio and an enhanced vertical mixing. Black lines denote contours of DO concentrations of 20 mmol m⁻³ and 60 mmol m⁻³ from

WOA2013 data.

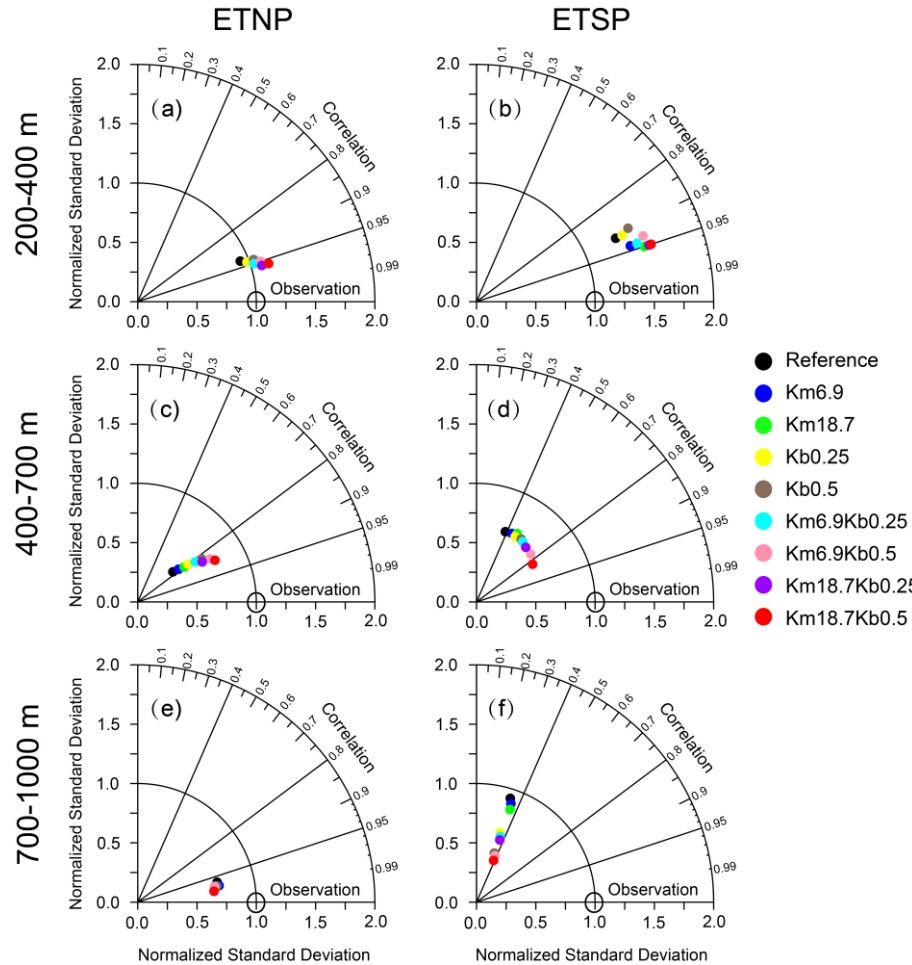

**Figure 5.** Taylor diagrams for the performance of simulated DO concentration (against WOA2013) from model simulations for ETNP (165°W-90°W, 5°N-20°N, left panel) and ETSP (110°W-80°W, 10°S-3°S, right panel) over **(a and b)** 200-400 m, **(c and d)** 400-700 m, and **(e and f)** 700-1000 m.

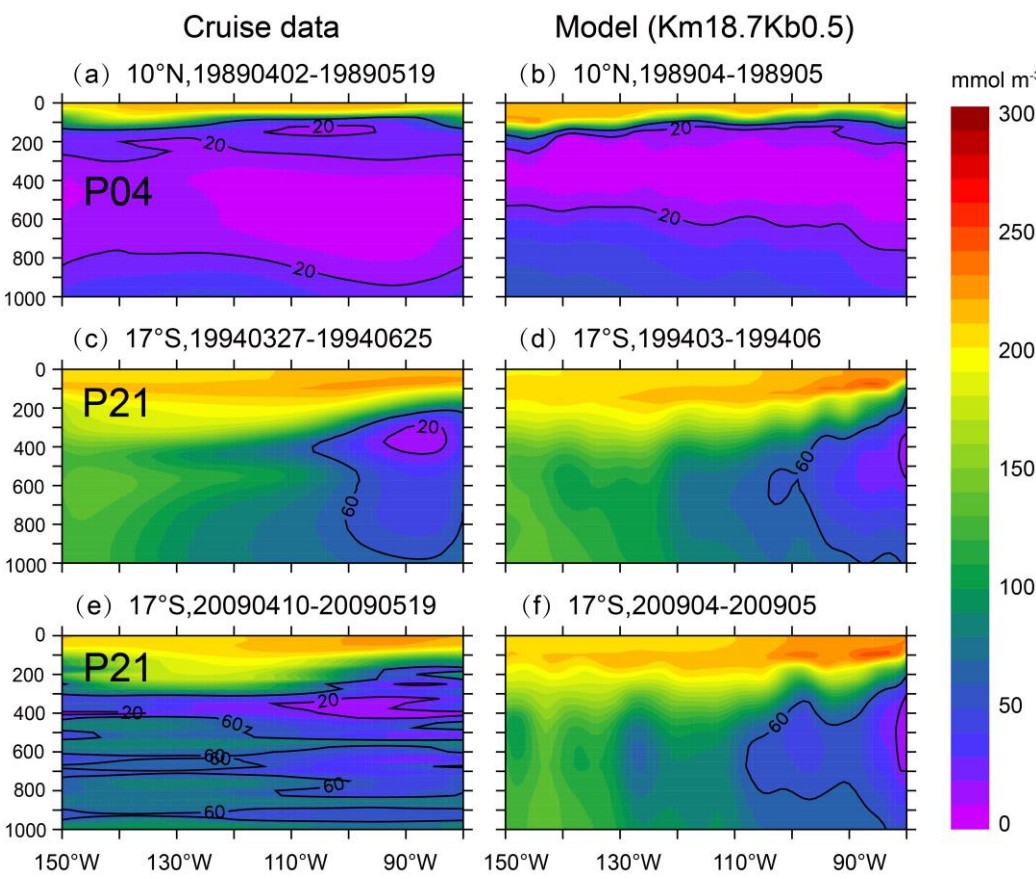


**Figure 6.** Distribution of DO from cruise data (left panel) and model simulation from Km18.7Kb0.5 (right panel). Observed DO data are from CCHDO (https://cchdo.ucsd.edu/), along **(a)** P04 (10°N) during April 02 - May 19, 1989, **(c)** P21 (17°S) during March 27 - June 25, 1994, and **(e)** P21 (17°S) during April 10 - May 19, 2009.


# 4 Model evaluation and discussions

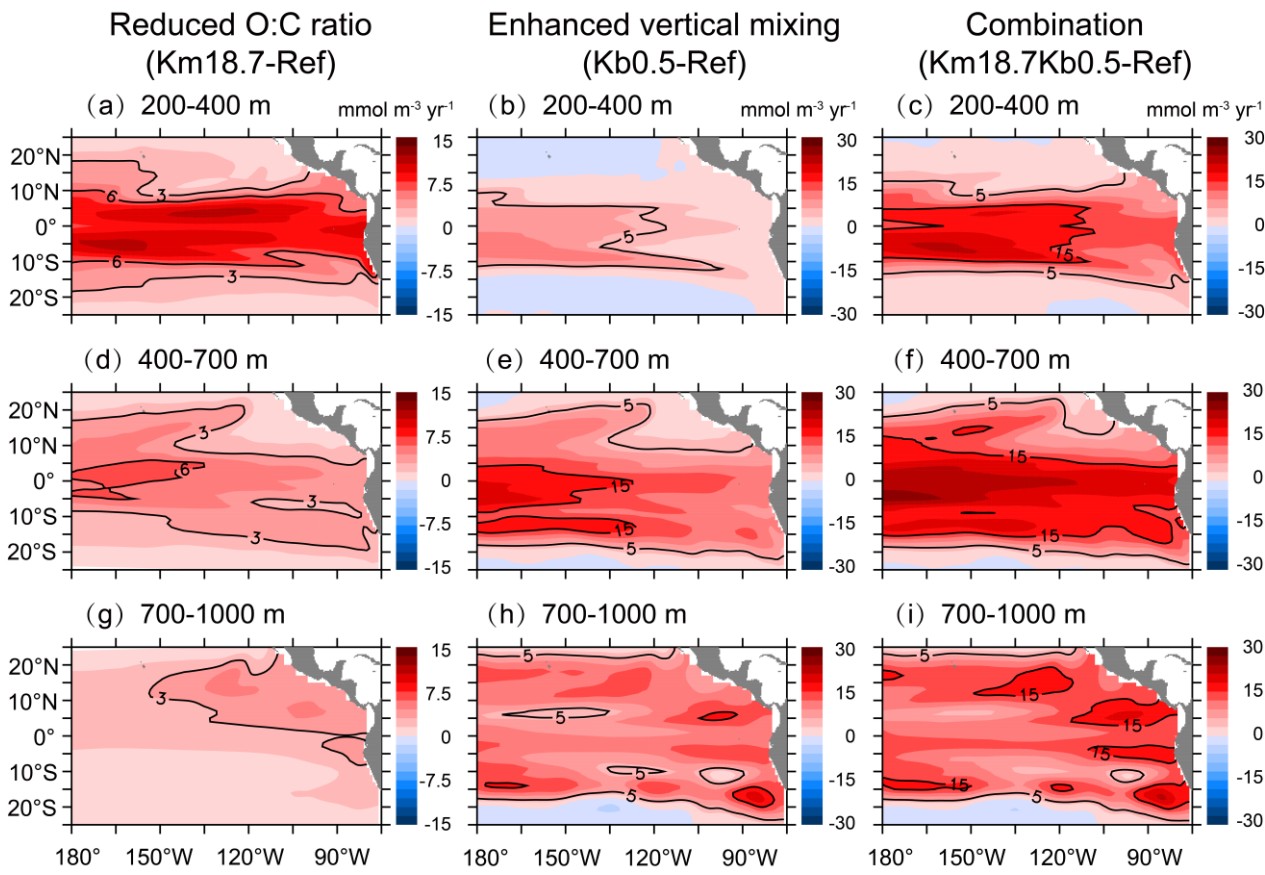

**Figure 7.** Changes in DO concentration over **(a, b and c)** 200-400 m, **(d, e and f)** 400-700 m, and **(g, h and i)** 700-1000 m due to a reduced O:C utilization ratio (left panel), an enhanced vertical mixing (middle panel), and the combination of a reduced O:C utilization ratio and an enhanced vertical mixing (right panel).

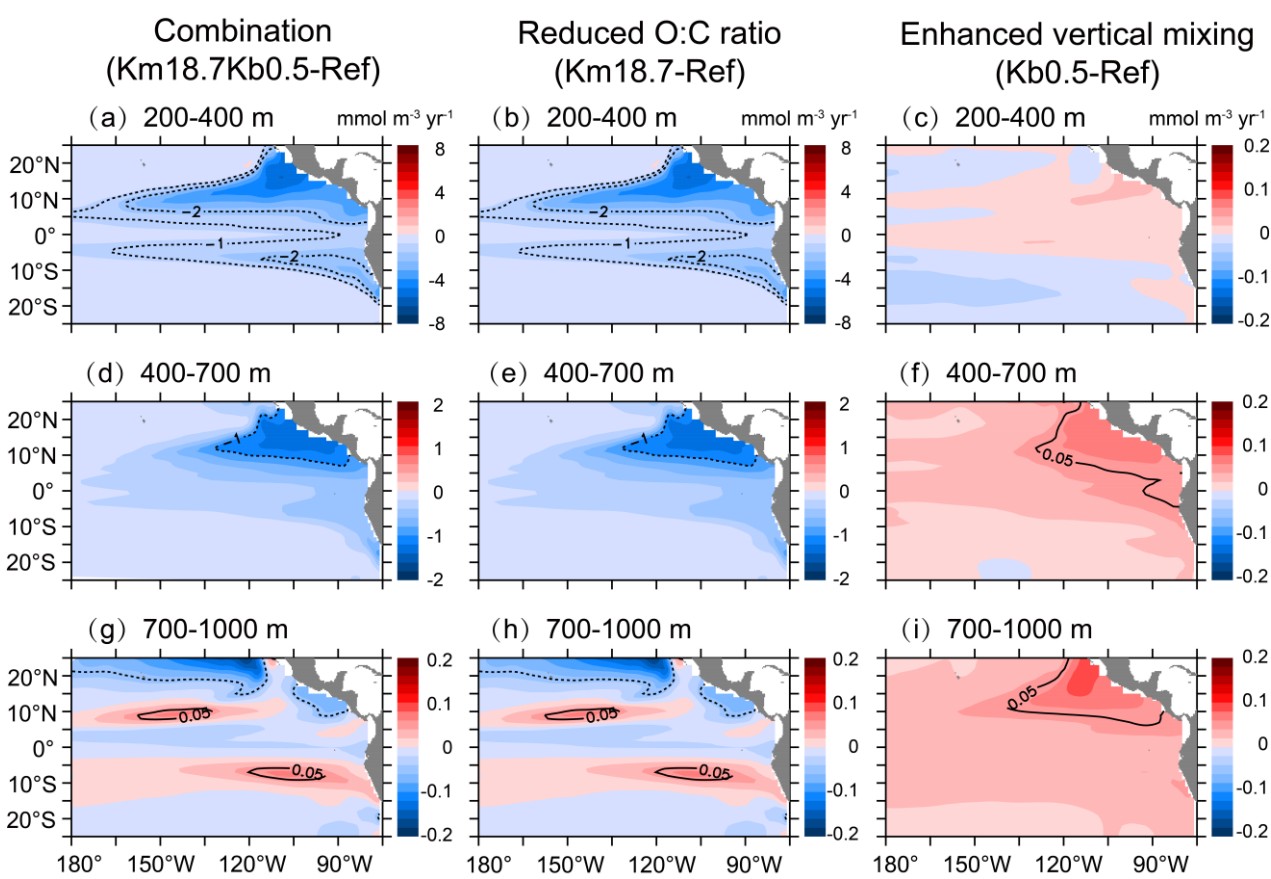


**Figure 8.** Changes in biological consumption over **(a, b and c)** 200-400 m, **(d, e and f)** 400-700 m, and **(g, h and i)** 700-1000 m due to the combination of a reduced O:C utilization ratio and an enhanced vertical mixing (left panel), a reduced O:C utilization ratio (middle panel), and an enhanced vertical mixing (right panel).


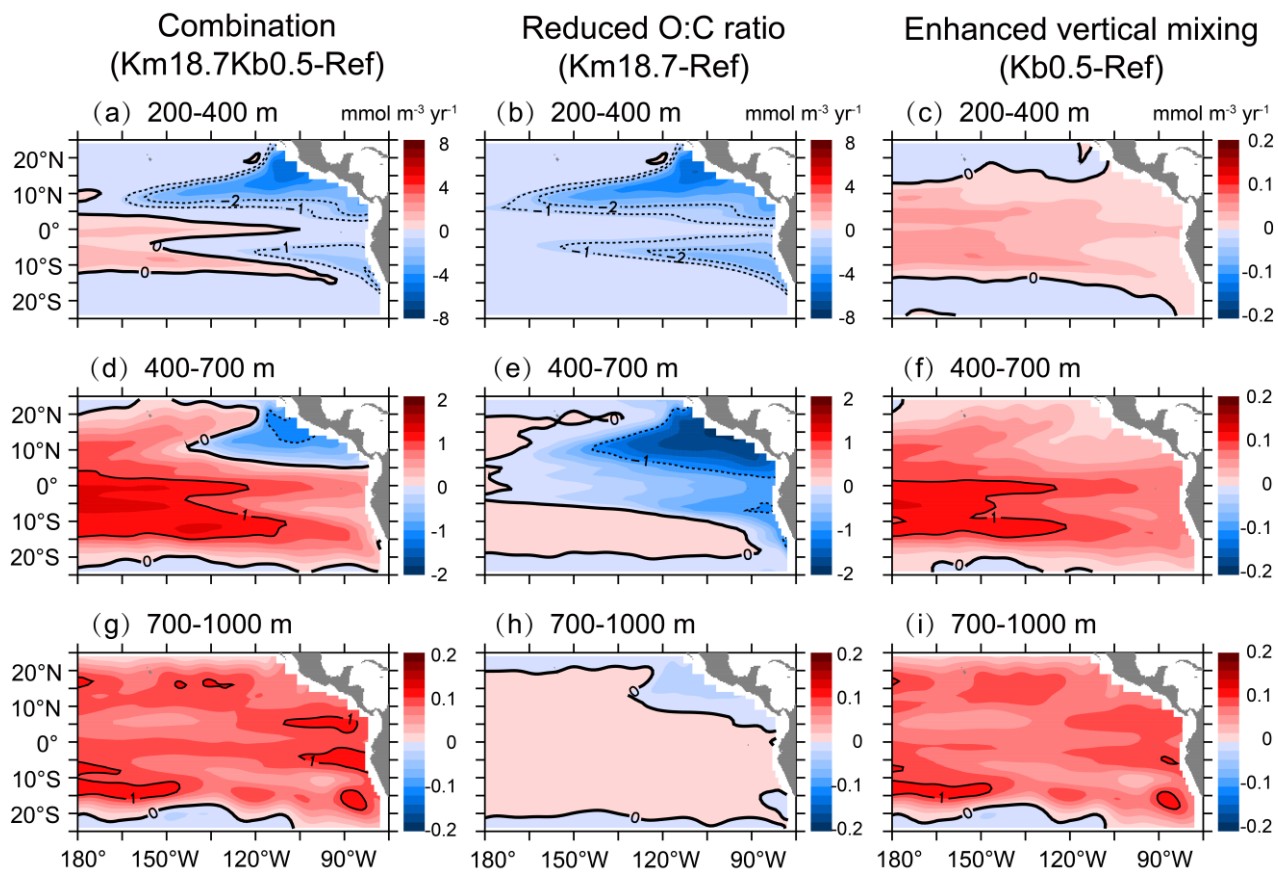

**Figure 9.** Changes in physical supply over **(a, b and c)** 200-400 m, **(d, e and f)** 400-700 m, and **(g, h and i)** 700-1000 m due to the combination of a reduced O:C utilization ratio and an enhanced vertical mixing (left panel), a reduced O:C utilization ratio (middle panel), and an enhanced vertical mixing (right panel).

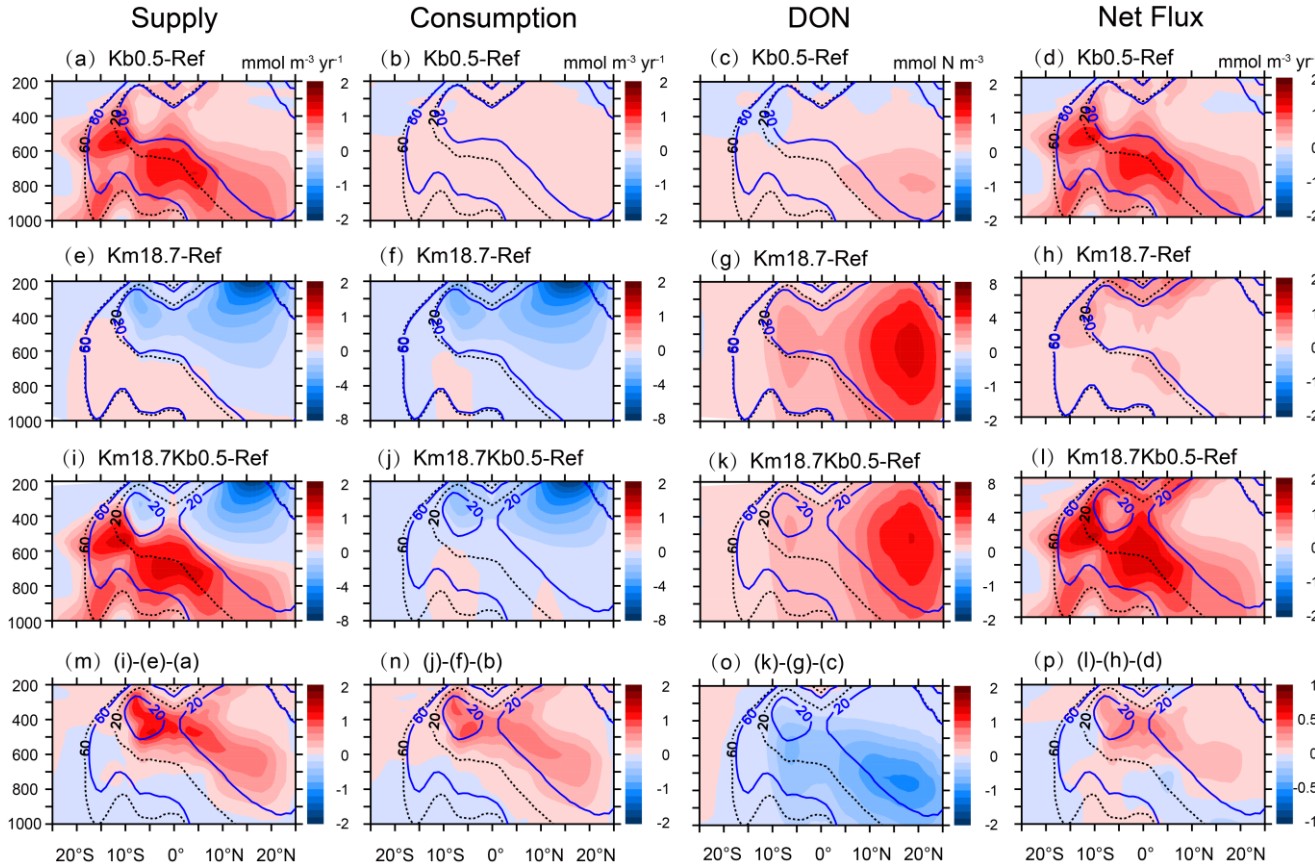

**Figure 10.** Changes in physical supply, biological consumption, DON concentration and net flux over 120°W-90°W under an enhanced vertical mixing **(a, b, c, and d, top row)**, a reduced O:C utilization ratio **(e, f, g, and h, second row)**, combination effects **(i, j, k, and l, third row)**, and the residuals or interactions (**m, n, o, and p, bottom row**). Black and blue lines denote contours of DO concentrations of 20 mmol m⁻³ and 60 mmol m⁻³ from the reference run (dashed lines) and other simulations (solid lines).

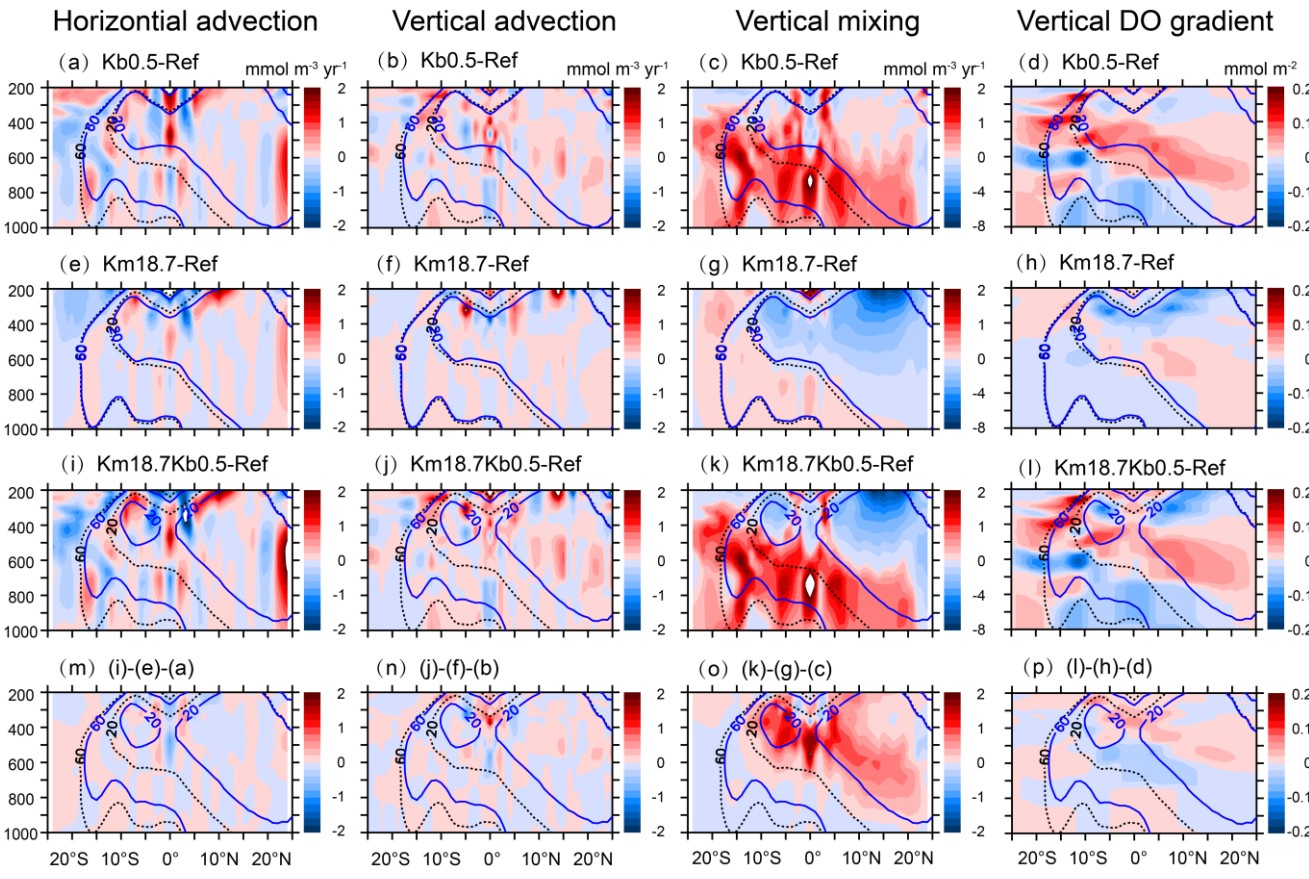


**Figure 11.** Changes in horizontal advection, vertical advection, vertical mixing and vertical DO gradient over 120°W-90°W under an enhanced vertical mixing (**a, b, c, and d, top row**), a reduced O:C utilization ratio (**e, f, g, and h, second row**), and combination effects (**i, j, k, and l, third row**), and the residues or interactions (**m, n, o, and p, bottom row**). Black and blue lines denote contours of DO concentrations of 20 mmol m$^{-3}$ and 60 mmol m$^{-3}$ from the reference run (dashed lines) and other simulations (solid lines).