# Peer review of "Sensitivity of asymmetric Oxygen Minimum Zones to mixing intensity and stoichiometry in the tropical Pacific using a basin-scale model (OGCM-DMEC V1.4)"

_Geoscientific Model Development, 2020_

## Author Comment (AC1)

Thanks for the constructive comments. Since this is a model evaluation paper (not model development paper), we only provide a brief description on OGCM and biogeochemical model. But, we could add some details particularly for the biogeochemical model if space is allowed. Oxygen has been a state variable (just like carbon and nitrogen) in the basin-scale biogeochemical model. Most parameters used to compute the sources/sinks of oxygen are the same as those for nitrogen and carbon cycles. We analyzed/validated many biogeochemical variables in our previous studies, e.g., chlorophyll (Wang *et al.*, 2009a; Wang *et al.*, 2013), nitrogen uptake and regeneration (Wang *et al.*, 2009b) and carbon cycling (Wang *et al.*, 2006b; Wang *et al.*, 2015). In addition, we reported/validated PP & NCP (Wang *et al.*, 2006b), new production (Wang *et al.*, 2006a), and nitrate, iron, POC/detritus and export production below 200 m (Yu *et al.*, 2021). Although this is the first manuscript reporting mode calibration and validation for oxygen cycle, we have presented some findings on oxygen cycle modeling at a few international conferences, e.g., AGU Fall Meeting 2016, AOGS Annual Meeting 2016, SFB754 (2018) and 2018 Climate Change Symposium.

Wang, X. J., Behrenfeld, M., Le Borgne, R., Murtugudde, R. & Boss, E. (2009a). Regulation of phytoplankton carbon to chlorophyll ratio by light, nutrients and temperature in the Equatorial Pacific Ocean: a basin-scale model. *Biogeosciences* 6(3): 391-404.

Wang, X. J., Christian, J. R., Murtugudde, R. & Busalacchi, A. J. (2006a). Spatial and temporal variability in new production in the equatorial Pacific during 1980-2003: Physical and biogeochemical controls. *Deep-Sea Research Part II* 53(5-7): 677-697.

Wang, X. J., Christian, J. R., Murtugudde, R. & Busalacchi, A. J. (2006b). Spatial and temporal variability of the surface water pCO(2) and air-sea CO2 flux in the equatorial Pacific during 1980-2003: a basin-scale carbon cycle model. *Journal of Geophysical Research* 111(C7): C07S04, doi:10.1029/2005JC002972.

Wang, X. J., Le Borgne, R. & Murtugudde, R. (2009b). Nitrogen uptake and regeneration pathways in the equatorial Pacific: a basin scale modeling study. *Biogeosciences* 6: 2647-2660.

Wang, X. J., Murtugudde, R., Hackert, E., Wang, J. & Beauchamp, J. (2015). Seasonal to decadal variations of sea surface pCO(2) and sea-air CO2 flux in the equatorial oceans over 1984-2013: A basin-scale comparison of the Pacific and Atlantic Oceans. *Global Biogeochemical Cycles* 29(5): 597-609.

Wang, X. J., MurtuguddeA, R., Hackert, E. & Maranon, E. (2013). Phytoplankton carbon and chlorophyll distributions in the equatorial Pacific and Atlantic: A basin-scale comparative study. *Journal of Marine Systems* 109: 138-148.

Yu, J., Wang, X., Murtugudde, R., Tian, F. & Zhang, R.-H. (2021). Interannual-to-Decadal Variations of Particulate Organic Carbon and the Contribution of Phytoplankton in the Tropical Pacific During 1981–2016: A

Model Study. *Journal of Geophysical Research: Oceans* 126(1): e2020JC016515.

---

## Author Comment (AC2)

Comment on gmd-2020-431

Anonymous Referee #1

Referee comment on "Sensitivity of asymmetric Oxygen Minimum Zones to remineralization rate and mixing intensity in the tropical Pacific using a basin-scale model (OGCM-DMEC V1.2)" by Kai Wang et al., Geosci. Model Dev. Discuss., https://doi.org/10.5194/gmd-2020-431-RC1, 2021

General comments:

Wang and co-workers address in their paper "Sensitivity of asymmetric Oxygen Minimum Zones to remineralization rate and mixing intensity in the tropical Pacific using a basinscale model (OGCM-DMEC V1.2)" one of the still open issues on understanding the interplay between the physical ocean and the marine biogeochemistry in shaping OMZs. Based on a basin-scale model with a high horizontal resolution they perform sensitivity studies with a set of vertical mixing parameters and a reduced DON remineralisation rate. With the final parameter set they state that the model successfully reproduces the observed asymmetric OMZ. Unfortunately, there is no new scientific finding in this paper. The results are very descriptive without any critical assessment. Furthermore, the "improved model" setup is only evaluated wrt. oxygen distributions. Potential effects on other biological components and/or processes due to the new parametrisation are not analysed, even so, changes in the OMZ might feedback onto the net community production. The authors state in their conclusion that a "reduced remineralization rate leads to remarkable decrease of biological consumption over 200- 400 m". This is a rather trivial finding.

**Reply**: Thanks for the constructive comments. We will make major revisions by considering all reviewers' comments and suggestions.

The physical and biological component of the applied OGCM are not sufficiently introduced. Only after reading previous papers of the authors I could gain a rudimentary understanding of the physical model setup. It would be useful to describe at least the major characteristics of the physical model. I also find that it is not a sufficient introduction of the biogeochemical component to only provide its equations in an Appendix. A more detailed description might be "boring" to the authors but it is very useful for the readers to get the basic concept of their model assumptions. Moreover the oxygen cycle, the core topic of this paper, seems to be newly implemented into the biogeochemical module. However, a comprehensive introduction is missing. It would be interesting to know: 1) how is guaranteed that oxygen consumption does not exceed available oxygen? 2) are there any restrictions to

remineralization depending on oxygen levels? Oxygen consumption from NH4 oxidation seems to be missing or is neglected. Oxygen production/consumption is calculated with a fixed ratio from NCP. However, photosynthesis based on nitrate produces a higher amount of oxygen than on NH4. Similarily, remineralization to NH4 needs less O2.

**Reply**: Thanks for the constructive comments. Since this is a model evaluation paper (not model development paper), we only provide a brief description on the OGCM and biogeochemical model. But, we could add some details particularly for the biogeochemical model if space is allowed. Dissolved oxygen (DO) has been a state variable (just like dissolved inorganic carbon) in the basin-scale biogeochemical model. Most parameters used to compute the sources/sinks of oxygen are the same as those used to compute the sources/sinks of nitrogen and carbon. We have analyzed/validated many biogeochemical variables in our previous studies, e.g., chlorophyll (Wang *et al.*, 2009a; Wang *et al.*, 2013), nitrogen uptake and regeneration (Wang *et al.*, 2009b) and carbon cycling (Wang *et al.*, 2006b; Wang *et al.*, 2015).

Regarding "1) how is guaranteed that oxygen consumption does not exceed available oxygen? 2) are there any restrictions to remineralization depending on oxygen levels?", we have been testing the parameterization with sensitivity experiments. Some studies have suggested that remineralization of DOM is retarded by low-DO conditions (Kalvelage *et al.*, 2015; Laufkötter *et al.*, 2017), which yield a wide range of half-saturation constant. We will include the sensitivity experiments in the revised manuscript.

The model does include "oxygen consumption from NH4 oxidation" (see eq. B9), which is a portion of $c_{DON}DON + c_{DS}D_s + c_{DL}D_L$. The impacts of nitrogen cycle on oxygen consumption/production or the interactions between oxygen cycle and nitrogen cycle are complex, according to some studies (Sun *et al.*, 2021; Kalvelage *et al.*, 2013). But there is limited information available for the parameterizations of relevant processes. In order to model these processes, we need more field data that allow us to quantify oxygen consumption/production in associated processes and to derive relevant parameters.

Furthermore, there is no sentence on nitrogen reduction processes such as denitrification, whose activity is highly correlated with export production in the ETSP (Kalvelage et al, 2013). As no values are given for NCP, export production, and also the distributions of nutrients or detritus are not provided it is impossible for the reader to judge the quality of the model performance. As far as I know, there has been previously no assessment of their biological component for the depth range below 200m. In view of the extensive degree of revision, I refrain from specific and technical comments.

**Reply**: Thanks for the constructive comments. Because there is limited information available for calibration and validation of some nitrogen reduction processes including denitrification, the regional model does not simulate them at this stage but will include these processes in the future once information becomes available. We have reported/validated many biogeochemical fields, including PP & NCP (Wang *et al.*, 2006b), new production (Wang *et al.*, 2006a) in the euphotic zone, and nitrate, iron, POC/detritus and export production below 200 m (Yu *et al.*, 2021). We will add some of the references in the revised manuscript.

**Reference**

Kalvelage, T., Lavik, G., Jensen, M. M., Revsbech, N. P., Loscher, C., Schunck, H., Desai, D. K., Hauss, H., Kiko, R., Holtappels, M., LaRoche, J., Schmitz, R. A., Graco, M. I. & Kuypers, M. M. (2015). Aerobic microbial respiration In oceanic oxygen minimum zones. *PLoS One* 10(7).

Kalvelage, T., Lavik, G., Lam, P., Contreras, S., Arteaga, L., Löscher, C. R., Oschlies, A., Paulmier, A., Stramma, L. & Kuypers, M. M. M. (2013). Nitrogen cycling driven by organic matter export in the South Pacific oxygen minimum zone. *Nature Geoscience* 6(3): 228-234.

Laufkötter, C., John, J. G., Stock, C. A. & Dunne, J. P. (2017). Temperature and oxygen dependence of the remineralization of organic matter. *Global Biogeochemical Cycles* 31(7): 1038-1050.

Sun, X., Frey, C., Garcia-Robledo, E., Jayakumar, A. & Ward, B. B. (2021). Microbial niche differentiation explains nitrite oxidation in marine oxygen minimum zones. *ISME J* 15(5): 1317-1329.

Wang, X. J., Behrenfeld, M., Le Borgne, R., Murtugudde, R. & Boss, E. (2009a). Regulation of phytoplankton carbon to chlorophyll ratio by light, nutrients and temperature in the Equatorial Pacific Ocean: a basin-scale model. *Biogeosciences* 6(3): 391-404.

Wang, X. J., Christian, J. R., Murtugudde, R. & Busalacchi, A. J. (2006a). Spatial and temporal variability in new production in the equatorial Pacific during 1980-2003: Physical and biogeochemical controls. *Deep-Sea Research Part II* 53(5-7): 677-697.

Wang, X. J., Christian, J. R., Murtugudde, R. & Busalacchi, A. J. (2006b). Spatial and temporal variability of the surface water pCO(2) and air-sea CO2 flux in the equatorial Pacific during 1980-2003: a basin-scale carbon cycle model. *Journal of Geophysical Research* 111(C7): C07S04, doi:10.1029/2005JC002972.

Wang, X. J., Le Borgne, R. & Murtugudde, R. (2009b). Nitrogen uptake and regeneration pathways in the equatorial Pacific: a basin scale modeling study. *Biogeosciences* 6: 2647-2660.

Wang, X. J., Murtugudde, R., Hackert, E., Wang, J. & Beauchamp, J. (2015). Seasonal to decadal variations of sea surface $pCO_2$ and sea-air $CO_2$ flux in the equatorial oceans over 1984-2013: A basin-scale comparison of the Pacific and Atlantic Oceans. *Global Biogeochemical Cycles* 29(5): 597-609.

Wang, X. J., MurtuguddeA, R., Hackert, E. & Maranon, E. (2013). Phytoplankton carbon and chlorophyll distributions in the equatorial Pacific and Atlantic: A basin-scale comparative study. *Journal of Marine Systems* 109: 138-148.

Yu, J., Wang, X., Murtugudde, R., Tian, F. & Zhang, R.-H. (2021). Interannual-to-Decadal Variations of Particulate Organic Carbon and the Contribution of Phytoplankton in the Tropical Pacific During 1981–2016: A Model Study. *Journal of Geophysical Research: Oceans* 126(1): e2020JC016515.

---

## Author Comment (AC3)

Referee comment on "Sensitivity of asymmetric Oxygen Minimum Zones to remineralization rate and mixing intensity in the tropical Pacific using a basin-scale model (OGCM-DMEC V1.2)" by Kai Wang et al., Geosci. Model Dev. Discuss., https://doi.org/10.5194/gmd-2020-431-RC2, 2021

The manuscript, "Sensitivity of asymmetric Oxygen Minimum Zones to remineralization rate and mixing intensity in the tropical Pacific using a basin-scale model (OGCM-DMEC V1.2)" by Wang et al. conduct a suite of model parameter sensitivity experiments with a very old, coarse resolution regional physical ocean model. While using an older model is not necessarily a disadvantage, it is only an advantage is the relative strengths and weaknesses of the model are provided such that the reader can integrate the current analysis to other current understanding. That context is not currently provided. For example, focusing on this region has the advantage that the sponge resets the source O2 (a major weakness of global models) to observations (the authors should note this strength of the current approach). Unfortunately, the comparability of the physical formulation to other models is missing. For example, it is unclear whether the Indonesian Throughflow is represented which is an important part of the advective ventilation in the Western part of the basin and the partitioning of lateral oxygen source waters into the Eastern part of the basin.

**Reply**: Thanks for the constructive comments. In this regional configuration, the model closes the western boundary and no representation of the Indonesian throughflow is included. Consistent with our previous publications (Wang et al., 2008; Wang et al., 2015; Yu et al., 2021) and numerous other studies (Duteil, 2019; Duteil et al., 2020; Ito and Deutsch, 2013; Llanillo et al., 2018) which focus on the tropical Pacific without the ITF, we rely on the imposed meridional boundary relaxation to constrain our regional solution. Clearly this is inadequate in the strictest sense of the processes mentioned by the reviewer. We posit that the validations presented support our contention that the model is reasonably constrained for the timescales we are considering in this study. Further studies will include the Indonesian-Pacific configuration with an explicit representation of the ITF and the O2 ventilation into the domain as reported in Rodgers et al. (1999). The current focus is on the Pacific processes which we deem are adequately represented in the current configuration. We will provide more details on the model, including configuration, boundary conditions and so on.

The analysis uses an inappropriate definition of "suboxic" (see below). Throughout the manuscript the word "rates" is used when "rate constant" is intended (e.g. on line 204 "Reducing remineralization rate by 50% (Cd0.5 minus reference) leads to large decrease…") making it difficult to interpret the result since it is unclear whether the "rate" is proportionally reduced by 50% with fixed concentration or whether there are compensating responses/increases in concentration that result in a change in the remineralization locations. While the result of the combined need to reduce the remineralization rate constant and increase the vertical diffusivity to better match oxygen distributions is encouraging, the manuscript oddly stops there without coming to any implications of the work for our understanding of the oxygen and nitrogen cycles or the past or future of the OMZ. What was

learned that wasn't known before? Most importantly, the final sentence of the conclusions, "Future studies utilizing advanced models are needed to better understand the impacts of physical and biological interactions on the variability and drivers of the tropical OMZs." Suggests the authors themselves are unclear as to the significance of the present work to current ocean biogeochemical modeling. As such, I recommend the authors work to clarify there descriptions and the implications and limitations of the current work in revision.

**Reply**: Thanks for the constructive comments. Regarding the definition of "suboxic", previous studies have used a wide range of DO as a criterion, e.g., <5 mmol $m^{-3}$ (Bianchi et al., 2012; Karstensen et al., 2008; Yakushev and Neretin, 1997) and <20 mmol $m^{-3}$ (Babbin et al., 2015; Helly and Levin, 2004; Oguz et al., 2000; Wright et al., 2012). Some researchers selected DO < 20 mmol $m^{-3}$ as the boundary of OMZs (Bettencourt et al., 2015; Fuenzalida et al., 2009; Paulmier and Ruiz-Pino, 2009). Accordingly, we adopt the criterion of <20 mmol $m^{-3}$ for both suboxic water and OMZ volume.

"Reducing remineralization rate by 50%" means applying reduced constant ($C_{DON}$) of remineralization (by 50%, i.e., Cd0.5). We will make necessary changes to clarify this, and also improve the interpretation with more in-depth analyses. We will take into consideration of all reviewers' comments and suggestions during the major revisions, including "implications of the work for our understanding of the oxygen and nitrogen cycles".

Technical comments:

Line 26 –"which made significant progresses" needs rephrasing.

**Reply**: we will rephrase the sentence.

Line 40 – The authors are misinformed as to the definition of "suboxic", quoting a value of 20 mmol m-3… suboxia is defined as an oxygen level at or below the detection limit, typically 2-10 mmol m-3 where interesting nitrogen redox chemistry such as N2O production, denitrification and annamox occur. The current definition of <20 is rather "strongly hypoxic" as it is well within the detectible range and well above the region of interesting redox chemistry. I would note that the reference the authors cite, Paulmier and Ruiz-Pinu (2009), use a suboxic level of 4.5 umol/gk. Also, if the authors want to describe the truly "suboxic" volume, they should be aware that while Table 3 notes a volume of "suboxic" waters from WOA13, it has been demonstrated that these mapped products strongly underestimate the volume of suboxia at the <5 mmol/m3 definition (Bianchi et al., 2012; https://agupubs.onlinelibrary.wiley.com/doi/full/10.1029/2011GB004209)

**Reply**: Thanks for the constructive comments. There is a wide range of DO value used to define suboxic water. In the paper of Paulmier and Ruiz-Pino (2009), they cited DO <4.5 umol/kg from an earlier study (Karstensen et al., 2008), but also stated "DO <20 umol/kg corresponds to a usual suboxic condition used to separate the aerobic (O2-respiration) from the denitrifying (NO3-respiration) activity (Oguz et al., 2000)". In addition, Wright et al. (2012) also defined 0-20 umol/kg as suboxic water (See Figure below). We will add some explanation regarding the definition of "suboxic" in the revised manuscript.

We were aware that Fuenzalida et al. (2009) and Bianchi et al. (2012) used the WOA2005 to estimate OMZ volume, and reported similar values at <20 mmol m$^{-3}$ definition. In our study, we derived similar OMZ volume ($5.97*10^6$ km$^3$ to the north and $1.43*10^6$ km$^3$ to the south) using WOA2013. We will mention the underestimation at <5 mmol m$^{-3}$ definition during the revision.

[Figure]

Figure 2 | O$_2$ **concentration affects ecosystem energy flow.** Alternative states of

Line 45-47 – There is an underlying assertion here that data alone provides understanding, and that more availability of data will resolve the underlying mechanisms. This is a false premise. Only by contextualizing the observations in a theoretical framework can mechanistic understanding be achieved. Also, "our understanding is uncompleted in terms" should be rephrased.

   **Reply**: We will rephrase these sentences.

Line 54 – "often" seems unnecessary here given that if the OMZ stretches across the equator it would seem to always lead to an overestimate of the OMZ area… unless there is a concomitant decline in area elsewhere in some models. If the latter is indeed the case, it would be worth mentioning. If the intent is just to point out the overestimate, then remove "often".

   **Reply**: We will remove "often".

Line 57 – "Apparently, it's necessary to…" this is an odd way of saying this, making it sound like the authors are annoyed at the idea.

   **Reply**: We will remove "Apparently".

Line 73 – This is a really old, coarse resolution model. A lot of advance has occurred over the last 25 years.

**Reply**: While this is an old model, we would not think it is a coarse resolution model given that the meridional resolution is 0.33º over 5ºS-5ºN. And the reviewer one called this model a high horizontal resolution model. Our previous studies have shown that this model can reproduce mesoscale and sub-mesoscale structures such as the tropical instability wave (TIW) (Tian et al., 2018; Zhang et al., 2018), and spatial and temporal variations of biogeochemical fields in the tropical Pacific (Wang et al., 2008; Wang et al., 2015).

Line 74-75 – What is the vertical grid? The stated 10-50m +20*10m layers = 210-250 m… this is not deep enough to represent the OMZ…?

**Reply**: The vertical resolution varies over depth, with ~30-50 m in the core OMZ (at 300-500 m) and the total depth of ~1200 m. We will make some rewording to clarify this.

Line 75 – What is the longitudinal grid? 150W-80E? Are the walls open to admit the Indonesian throughflow? This would seem critical for representation of O2 ventilation flow into the domain (e.g. Rodgers et al, 1999; https://agupubs.onlinelibrary.wiley.com/doi/abs/10.1029/1998JC900094). Is the Indonesian throughflow prescribed? Both factors should also be explicit.

**Reply**: The longitude of our model is from 124°E to 76°W. In this basin-scale model, the western boundary is closed, without representation of the Indonesian throughflow. Consistent with our previous publications and numerous other studies which focus on the tropical Pacific without the ITF, we rely on the imposed meridional boundary relaxation to constrain our regional solution. We posit that the validations presented support our contention that the model is reasonably constrained for the timescales we are considering in this study.

Line 96 – the parameters here described as "rates" are actually "rate constants", e.g. r is the rate constant for zooplankton respiration.

**Reply**: We will replace "rate" with "constant".

Line 116 – What does "DON poor" mean?

**Reply**: We will correct as "DON pool".

Line 128 – This implies that the model domain extends to 1000 m or more, suggesting line 74-75 is incorrect.

**Reply**: Yes, the model domain extends to ~1200 m.

Line 140 – It is important to note that "underestimation of supply" is complex and can be from either O2 being too low in the waters that supply or the physical supply mechanisms being either too sluggish or out of balance (e.g. lateral versus vertical and advective versus diffusive"

**Reply**: Thank you for the constructive comments. We agree that "underestimation of supply" is complex, and will make revision to address this issue, e.g., by adding new analyses such as advection, upwelling and diffusion of DO.

Line 145-149 – How did these perturbations influence the fidelity of T and S?

**Reply**: Enhanced vertical mixing (i.e., addition of background diffusion) only applies to the most important variables in this study (i.e., DO and DON). Thus, T and S are not influenced. We will clarify this in the revised manuscript.

Line 152 – What is the reference value of Cd? What does it do? There is no parameter called "Cd" is the appendix, only "CDON0" the remineralization rate constant at 10 C, but it's reference value, 0.001, is very different from 0.5. Looking at Table 1, I see that "Cd05" is actually "CDON0*0.5". However, it is not clear what the 100-600 m range of "0.0005-0.00025" means… is this the role of temperature on CDON0? This parameter needs a sentence or two of introduction, definition, and contextualization here to avoid confusion.

**Reply**: Yes, "Cd0.5" is actually "CDON0*0.5". We will provide more information for the model experiments, and also add a couple of sentences to clarify how CDON0 varies over depth.

Line 204 - the word "rates" is used when "rate constant" is intended (e.g. on line 204 "Reducing remineralization rate by 50% (Cd0.5 minus reference) leads to large decrease…") making it difficult to interpret the result since it is unclear whether the "rate" is proportionally reduced by 50% with fixed concentration or whether there are compensating responses/increases in concentration that result in a change in the remineralization locations.

**Reply**: We will add more details on model experiments, and rephrase some relevant definition/statements.

Line 211 – "there is somehow a small decrease…" The use of "somehow" is an insufficient explanation… what is causing this decrease? Is it a response to the remineralization constant decrease?

**Reply**: We will rephrase that sentence (regarding "somehow"). The decrease in supply with enhanced vertical mixing was mainly a result of change in vertical distribution of DO, caused by enhanced downward transportation of DON, which led to lower DON concentration in the OMZ and thus less DO consumption. The relevant discussion was in the next two paragraphs. We will rewrite some of the sentences to make this point clearer.

Line 224 – Only here is it explained that there was no response in temperature to the diffusivity change. This should have been noted earlier in the results as requested above, as well as the salinity response.

**Reply**: Yes, there was no response in temperature. We will address this point during the revision.

Line 228 – "Limited field studies" – why is the defining feature of these studies that they were "limited"? Is the evidence derived from them inconclusive? More explanation of context would be helpful.

**Reply**: We will phrase that sentence.

**Reference:**

Babbin, A. R., Bianchi, D., Jayakumar, A., and Ward, B. B.: Rapid nitrous oxide cycling in the suboxic ocean, Science, 348, 1127-1129, 2015.

Bettencourt, J. H., Lopez, C., Hernandez-Garcia, E., Montes, I., Sudre, J., Dewitte, B., Paulmier, A., and Garcon, V.: Boundaries of the Peruvian oxygen minimum zone shaped by coherent mesoscale dynamics, Nature Geoscience, 8, 937-U967, 2015.

Bianchi, D., Dunne, J. P., Sarmiento, J. L., and Galbraith, E. D.: Data-based estimates of suboxia, denitrification, and N2O production in the ocean and their sensitivities to dissolved O2, Global Biogeochemical Cycles, 26, 1-13, 2012.

Duteil, O.: Wind Synoptic Activity Increases Oxygen Levels in the Tropical Pacific Ocean, Geophysical Research Letters, 46, 2715-2725, 2019.

Duteil, O., Frenger, I., and Getzlaff, J.: Intermediate water masses, a major supplier of oxygen for the eastern tropical Pacific ocean, Ocean Science, doi: 10.5194/os-2020-17, 2020. 2020.

Fuenzalida, R., Schneider, W., Garces-Vargas, J., Bravo, L., and Lange, C.: Vertical and horizontal extension of the oxygen minimum zone in the eastern South Pacific Ocean, Deep-Sea Res Pt Ii, 56, 1027-1038, 2009.

Garcia, H. E., Locarnini, R. A., Boyer, T. P., Antonov, J. I., Mishonov, A. V., Baranova, O. K., Zweng, M. M., Reagan, J. R., and Johnson, D. R.: World Ocean Atlas 2013. Vol. 3: Dissolved Oxygen, Apparent Oxygen Utilization, and Oxygen Saturation, NOAA Atlas NESDIS, 75, 27, 2013.

Helly, J. J. and Levin, L. A.: Global distribution of naturally occurring marine hypoxia on continental margins, Deep-Sea Res Pt I, 51, 1159-1168, 2004.

Ito, T. and Deutsch, C.: Variability of the oxygen minimum zone in the tropical North Pacific during the late twentieth century, Global Biogeochemical Cycles, 27, 1119-1128, 2013.

Karstensen, J., Stramma, L., and Visbeck, M.: Oxygen minimum zones in the eastern tropical Atlantic and Pacific oceans, Progress in Oceanography, 77, 331-350, 2008.

Llanillo, P. J., Pelegri, J. L., Talley, L. D., Pena-Izquierdo, J., and Cordero, R. R.: Oxygen Pathways and Budget for the Eastern South Pacific Oxygen Minimum Zone, Journal of Geophysical Research: Oceans, 123, 1722-1744, 2018.

Oguz, T., Ducklow, H. W., and Malanotte-Rizzoli, P.: Modeling distinct vertical biogeochemical structure of the Black Sea: Dynamical coupling of the oxic, suboxic, and anoxic layers, Global Biogeochemical Cycles, 14, 1331-1352, 2000.

Paulmier, A. and Ruiz-Pino, D.: Oxygen minimum zones (OMZs) in the modern ocean, Progress in Oceanography, 80, 113-128, 2009.

Rodgers, K. B., Cane, M. A., Naik, N. H., and Schrag, D. P.: The role of the Indonesian Throughflow in equatorial Pacific thermocline ventilation, Journal of Geophysical Research: Oceans, 104, 20551-20570, 1999.

Tian, F., Zhang, R. H., and Wang, X.: A Coupled Ocean Physics‐Biology Modeling Study on Tropical Instability Wave‐Induced Chlorophyll Impacts in the Pacific, Journal of Geophysical Research: Oceans, 123, 5160-5179, 2018.

Wang, X. J., Le Borgne, R., Murtugudde, R., Busalacchi, A. J., and Behrenfeld, M.: Spatial and temporal variations in dissolved and particulate organic nitrogen in the equatorial Pacific: biological and physical influences, Biogeosciences, 5, 1705-1721, 2008.

Wang, X. J., Murtugudde, R., Hackert, E., Wang, J., and Beauchamp, J.: Seasonal to decadal variations of sea surface $pCO_2$ and sea-air $CO_2$ flux in the equatorial oceans over 1984–2013: A basin-scale comparison of the Pacific and Atlantic Oceans, Global Biogeochemical Cycles, 29, 597-609, 2015.

Wright, J. J., Konwar, K. M., and Hallam, S. J.: Microbial ecology of expanding oxygen minimum zones, Nature Reviews Microbiology, 10, 381-394, 2012.

Yakushev, E. V. and Neretin, L. N.: One-dimensional modeling of nitrogen and sulfur cycles in the aphotic zones of the Black and Arabian Seas, Global Biogeochemical Cycles, 11, 401-414, 1997.

Yu, J., Wang, X., Murtugudde, R., Tian, F., and Zhang, R. H.: Interannual‐to‐Decadal Variations of Particulate Organic Carbon and the Contribution of Phytoplankton in the Tropical Pacific During 1981–2016: A Model Study, Journal of Geophysical Research: Oceans, 126, 2021.

Zhang, R. H., Tian, F., and Wang, X. J.: A New Hybrid Coupled Model of Atmosphere, Ocean Physics, and Ocean Biogeochemistry to Represent Biogeophysical Feedback Effects in the Tropical Pacific, Journal of Advances in Modeling Earth Systems, 10, 1901-1923, 2018.

---

## Author Comment (AC4)

Referee comment on "Sensitivity of asymmetric Oxygen Minimum Zones to remineralization rate and mixing intensity in the tropical Pacific using a basin-scale model (OGCM-DMEC V1.2)" by Kai Wang et al., Geosci. Model Dev. Discuss., https://doi.org/10.5194/gmd-2020-431-RC3, 2021

This paper examines the sensitivity of the oxygen minimum zones (OMZs) to a change in the remineralization rate and changes in the vertical diffusion coefficient in the tropical Pacific. The goal of this study is to present a calibrated model, to evaluate this model and to identify the mechanisms that explain the asymmetric shape of the OMZ in the tropical Pacific.

Unfortunately this paper only shows the impact of two parameter changes, change in the remineralization rate and the vertical background diffusivity, mainly on the oxygen fields and DON distribution without providing an explanation about the driving mechanisms and a thorough discussion of the results. With this there is so far no scientifically new finding in the current state of the manuscript. In particular an explanation about the mechanisms that drive the asymmetric shape of the OMZs in the tropical Pacific, that is already present in the reference simulation, is missing due to the mainly descriptive nature of the paper. In addition the language that is used is in many places imprecise. E.g. there is no differentiation between vertical and horizontal transport processes as both are termed physical transport.

**Reply**:Thank you for the constructive comments. We agree that there is a need to improve our manuscript with more in-depth analyses and through discussion of the results, and with explanation about the mechanisms that drive the asymmetric shape of the OMZs in the tropical Pacific. We will make major revisions to address these issues. In particular, we will add more analyses on physical supply, including horizontal advection, and vertical transport, focusing on the differences between the north OMZ and south OMZ.

I think there is potential, that this model and the performed sensitivity simulations, could be used to explain the mechanisms that drive the asymmetry in the oxygen fields, although the current version leaves the reader with too many open questions. The model description is not sufficient. There is e.g. no information given about the biogeochemical boundary conditions. What does it mean: All biological components are computed in a manner similar to physical variables. What is the reason to average the model output for the period of 1981-2000 and do not include the last 18 years. Specially, as later in the manuscript some months of year 2009 are compared to some cruise data that are not introduced in the manuscript, they just appear.

Are the model data detrended before averaging? Is there a remaining model drift? Why not simply using climatological simulations? What kind of inter-annual forcing is used?

**Reply**:Thank you for the constructive comments. We will add more details in the model description section, including biogeochemical boundary conditions. "All biological components are computed in a manner similar to physical variables" means that all biological variables are influenced by physical processes (e.g., advection and diffusion). We will rephrase that sentence to clarify this.

We used model output from the period of 1981-2000 for the comparison with WOA2005 dataset because most oxygen data were obtained prior to 2000. Since we did not find much difference between WOA2005 and WOA2013, we did not change the period of 1981-2000 from model. But, we will use model outputs from 1981-2010 in the revised manuscript. We consider that it is necessary to use cruise data for further model validation, thus we did not using climatological simulations. We will provide more information about the datasets during the revisions.

The model data were not detrended before averaging, and we did not find a model drift. The inter-annual model forcing includes interannual 6-day means of precipitation (ftp://ftp.cdc.noaa.gov/Datasets/gpcp) and surface wind stress (from NCEP reanalysis, ftp://ftp.cdc.noaa.gov/Datasets/ncep.reanalysis2.dailyavgs).

Although it is mentioned at several places (abstract and introduction) that the model is calibrated - there is no further information given how. Is it calibrated only against the oxygen fields? As the reference simulation already represents the asymmetric shape of OMZ, I assume that this is at least partly a result of the model calibration. The only information that is in the paper: Some parameters have been changed compared to an earlier version of the model. This is however not sufficient.

**Reply**:Basin-scale model was calibrated against many biogeochemical fields, including chlorophyll (Wang et al., 2009a), nitrogen cycle (Wang et al., 2009b) and carbon cycle (Wang et al., 2015). In this study, we further calibrated against DON and oxygen fields. Although the reference simulation represents the asymmetric OMZ, it is not a result of the model calibration against biogeochemical fields. We believe that it is largely related to physical processes, e.g., sluggish circulation and weak ventilation associated with the sub-tropical gyres (Kuntz and Schrag, 2018; Wyrtki, 1962). We will add more information on model calibration and in-depth analyses of physical transport during the revision. We will also provide more details on model description, including parameterizations of relevant biogeochemical processes.

The model evaluation and validation is lacking. The oxygen fields look fine, but there is no information about, e.g. the circulation. How good is the current system represented in this model? What about nitrogen and phosphate - is there a bias in the east west gradient (one of the problems as shown e.g. by Dietze and Loeptien, 2013, doi:10.1002/gbc.20029) or is that reduced etc.

**Reply:** Our previous studies have provided model validation for chlorophyll (Wang et al., 2013), carbon fields (Wang et al., 2015; Yu et al., 2021). The model does a good job in representing the current system thus physical fields such as sea surface temperature (Wang et al., 2008; Zhang et al., 2018) (also see Figure a & c below). The model can also reproduce the west-to-east gradient of nitrate along the equator (see Figure b & d below). We will make improvement in model evaluation during the major revisions, aiming to better understand the responses of biological consumption and mid-depth DO to reduced remineralization and enhanced mixing, and relative roles of physical supply and biological consumption in shaping the asymmetric OMZs. We will also add new analyses on the regulation of advection and vertical mixing.

[Figure]

Regarding the structure of the paper - I would suggest to reorganize the paper: There should be a clear separation between the model set up as well as the set up of the sensitivity simulations and the results. In addition, regarding the comparison with the cruise data - these simply appear, where do they come from? Same happens with the DON data from HOT.

**Reply:** Thank you for your suggestion. We will make major revisions with some reorganization by taking into account all reviewers' comments and suggestions. In addition, we will make necessary changes in the sections of model description and/or model-data comparisons, with details about the cruise data (e.g., DO along P04 and P21,

https://cchdo.ucsd.edu) and station data (e.g., DON at HOT, https://hahana.soest.hawaii.edu/hot/hot_jgofs.html).

The sensitivity simulations show an improved representation of the OMZ in Fig4. The description of this improvement in the text is somewhat incorrect. The asymmetric shape is present in all simulations. It seems that the changes in the parameters result in an overall increase of DO and not necessarily an increase of the asymmetry. As at the southern hemisphere the DO concentrations are lower, one might get the impression that this increase might be slightly larger, but there is no evidence that this is the case at this stage. In addition, in the text it is stated that between 2°S-2°N the values of DO are relatively high (~30-40 mmol m-3) in Cd0.5Kb0.5. Fig 4f does not support this. Around 400m depth the DO concentrations are below 20 mmol m-3.

**Reply**:Thank you for the constructive comments. We get your point "not necessarily an increase of the asymmetry….", and realize the incorrect text (e.g., ~30-40 mmol m$^{-3}$). We will make necessary changes/corrections during the major revisions, not only in the description of Fig. 4, but also in the discussion of asymmetric OMZs and underlying mechanisms.

Fig 10 clearly shows differences - but there is no explanation given about the choice of the region shown as well as what have been done with the data - I guess they have been averaged. As the vertical oxygen gradients are different in these two regions, I would have expected a difference in the oxygen response, as the vertical diffusive transport depends on the gradient. Unfortunately the explanation of the results are again left to the reader.

**Reply**:Thank you for the constructive comments. The choice of the regions (ETNP and ETSP) is mainly for the suboxic water (DO <20 mmol m$^{-3}$), and all the values are means averaged for the box and over 1981-2000. We will add more details for Fig. 10, and also provide more explanation on the differences in oxygen responses between ETNP and ETSP, by assessing advection and vertical transport during the major revisions.

In addition there is no clear explanation given about the choice of the parameter change for the sensitivity simulations and with this it seems rather arbitrary. Also why is an increase of the vertical background diffusivity of 0.5 cm$^2$/s optimal, what about higher rates?

**Reply**:Thank you for the constructive comments. We will add explanation about the choice of the parameter during major revisions. In brief, new biological parameters were derived during model calibration against DON, and oxygen fields. Regarding the parameter for background diffusion, there is a large range (~0.01-0.5 cm$^2$/s) in other modeling studies

(Duteil and Oschlies, 2011; Ledwell et al., 1998; Matear and Hirst, 2003; Stramma et al., 2012); most studies reported improvements in oxygen at mid-depth using a value between 0.1 and 0.5 cm$^2$/s. While stronger vertical mixing could also lead to improved oxygen at mid-depth, a too-high value for background diffusion (mainly representing molecular diffusion) could not be accepted.

A thorough discussion of the results is missing. How are the results related to other modeling studies? The sensitivity studies show that the major changes occur along the equator - so this indicates that somehow the representation of the current system is important and needs to be shown and discussed. There are several physical processes in addition to the known impact of vertical mixing that seem to be capable to reduce the oxygen model bias along the equator, e.g. enhancing zonal diffusion or enhancing viscosity.

**Reply**:Thank you for the constructive comments. We will add more in-depth analyses (including comparisons with other modeling studies) with through discussion of the results. We will also provide more information on model validation, including physical fields (e.g., a good representation of the current system). We agree that other physical processes in addition to vertical mixing may be capable to reduce the oxygen model bias along the equator. But, this is beyond the scope of this study. Future studies will include sensitivity experiments with enhanced zonal diffusion or viscosity.

Also, as the model is forced with an inter-annually varying forcing, what about the potential impact of El Nino events?

**Reply**:Our preliminary analyses indicated that there were no significant impacts of El Nino events on mid-depth oxygen. But, El Nino events might have large impacts on oxygen fields in surface water. These analyses will be carried out in a future study.

As the manuscript needs substantial revisions, I do not add any specific and/or technical comments at this stage of the review process.

**Reference:**

Duteil, O. and Oschlies, A.: Sensitivity of simulated extent and future evolution of marine suboxia to mixing intensity, Geophysical Research Letters, 38, 2011.

Kuntz, L. B. and Schrag, D. P.: Hemispheric asymmetry in the ventilated thermocline of the Tropical Pacific, Journal of Climate 31, 1281-1288, 2018.

Ledwell, J. R., Watson, A. J., and Law, C. S.: Mixing of a tracer in the pycnocline, J Geophys Res-Oceans, 103, 21499-21529, 1998.

Matear, R. J. and Hirst, A. C.: Long-term changes in dissolved oxygen concentrations in the ocean caused by protracted global warming, Global Biogeochemical Cycles, 17, 2003.

Stramma, L., Oschlies, A., and Schmidtko, S.: Mismatch between observed and modeled trends in dissolved upper-ocean oxygen over the last 50 yr, Biogeosciences, 9, 4045-4057, 2012.

Wang, X. J., Behrenfeld, M., Le Borgne, R., Murtugudde, R., and Boss, E.: Regulation of phytoplankton carbon to chlorophyll ratio by light, nutrients and temperature in the Equatorial Pacific Ocean: a basin-scale model, Biogeosciences, 6, 391-404, 2009a.

Wang, X. J., Le Borgne, R., Murtugudde, R., Busalacchi, A. J., and Behrenfeld, M.: Spatial and temporal variations in dissolved and particulate organic nitrogen in the equatorial Pacific: biological and physical influences, Biogeosciences, 5, 1705-1721, 2008.

Wang, X. J., Murtugudde, R., Hackert, E., and Maranon, E.: Phytoplankton carbon and chlorophyll distributions in the equatorial Pacific and Atlantic: A basin-scale comparative study, Journal of Marine Systems, 109, 138-148, 2013.

Wang, X. J., Murtugudde, R., Hackert, E., Wang, J., and Beauchamp, J.: Seasonal to decadal variations of sea surface pCO2 and sea-air CO2 flux in the equatorial oceans over 1984–2013: A basin-scale comparison of the Pacific and Atlantic Oceans, Global Biogeochemical Cycles, 29, 597-609, 2015.

Wang, X. J., Murtugudde, R., and Le Borgne, R.: Nitrogen uptake and regeneration pathways in the equatorial Pacific: a basin scale modeling study, Biogeosciences, 6, 2647-2660, 2009b.

Wyrtki, K.: The oxygen minima in relation to ocean circulation, Deep Sea Research, 9, 11-23, 1962.

Yu, J., Wang, X., Murtugudde, R., Tian, F., and Zhang, R. H.: Interannual‐to‐Decadal Variations of Particulate Organic Carbon and the Contribution of Phytoplankton in the Tropical Pacific During 1981–2016: A Model Study, Journal of Geophysical Research: Oceans, 126, 2021.

Zhang, R. H., Tian, F., and Wang, X. J.: A New Hybrid Coupled Model of Atmosphere, Ocean Physics, and Ocean Biogeochemistry to Represent Biogeophysical Feedback Effects in the Tropical Pacific, Journal of Advances in Modeling Earth Systems, 10, 1901-1923, 2018.

---

## Author Response (AR1)

**Comment on gmd-2020-431**

**Anonymous Referee #1**

Referee comment on "Sensitivity of asymmetric Oxygen Minimum Zones to remineralization rate and mixing intensity in the tropical Pacific using a basin-scale model (OGCM-DMEC V1.2)" by Kai Wang et al., Geosci. Model Dev. Discuss., https://doi.org/10.5194/gmd-2020-431-RC1, 2021

**General comments:**

Wang and co-workers address in their paper "Sensitivity of asymmetric Oxygen Minimum Zones to remineralization rate and mixing intensity in the tropical Pacific using a basinscale model (OGCM-DMEC V1.2)" one of the still open issues on understanding the interplay between the physical ocean and the marine biogeochemistry in shaping OMZs. Based on a basin-scale model with a high horizontal resolution they perform sensitivity studies with a set of vertical mixing parameters and a reduced DON remineralisation rate. With the final parameter set they state that the model successfully reproduces the observed asymmetric OMZ. Unfortunately, there is no new scientific finding in this paper. The results are very descriptive without any critical assessment. Furthermore, the "improved model" setup is only evaluated wrt. oxygen distributions. Potential effects on other biological components and/or processes due to the new parametrisation are not analysed, even so, changes in the OMZ might feedback onto the net community production. The authors state in their conclusion that a "reduced remineralization rate leads to remarkable decrease of biological consumption over 200- 400 m". This is a rather trivial finding.

**Response**: Thanks for the constructive comments. We have made major revisions by taking into consideration of all reviewers' comments and suggestions. In particular, we have added more analyses on responses of biological consumption (equivalent to net community production in the OMZ) to the new parameterizations. In addition, we have assessed the responses of physical supply to changes in both physical and biological parameterizations, with in-depth analyses on the interactions/feedbacks and their impacts on the asymmetry of OMZs. We have rewritten the discussion and conclusion sections.

The physical and biological component of the applied OGCM are not sufficiently introduced. Only after reading previous papers of the authors I could gain a rudimentary understanding of the physical model setup. It would be useful to describe at least the major characteristics of the physical model. I also find that it is not a sufficient introduction of the biogeochemical component to only provide its equations in an Appendix. A more detailed description might be "boring" to the authors but it is very useful for the readers to get the basic concept of their model assumptions. Moreover the oxygen cycle, the core topic of this paper, seems to be newly implemented into the biogeochemical module. However, a comprehensive introduction is missing. It would be interesting to know: 1) how is guaranteed that oxygen consumption does not exceed available oxygen? 2) are there any restrictions to remineralization depending on oxygen levels? Oxygen consumption from NH4 oxidation seems to be missing or is neglected. Oxygen production/consumption is calculated with a fixed ratio from NCP. However, photosynthesis based on nitrate produces a higher amount of oxygen than on NH4. Similarily, remineralization to NH4 needs less O2.

**Response**: Thanks for the constructive comments. Since this is a model evaluation paper (not model development paper), we only provided a brief description on the OGCM and biogeochemical model in the previous version. But, we have added some details in the

revised manuscript, particularly for the biogeochemical model (line 104-107 and 113-120). Dissolved oxygen (DO) has been a state variable (just like dissolved inorganic carbon) in the basin-scale biogeochemical model. Most parameters used to compute the sources/sinks of oxygen are the same as those used to compute the sources/sinks of nitrogen and carbon. We have analyzed/validated many biogeochemical variables in our previous studies, e.g., chlorophyll (Wang *et al.*, 2009a; Wang *et al.*, 2013b), nitrogen uptake and regeneration (Wang *et al.*, 2009b) and carbon cycling (Wang *et al.*, 2015b).

Our sensitivity experiments include new simulations that have "restrictions to remineralization depending on oxygen levels", which guarantees oxygen consumption not exceeding available oxygen. The impacts of nitrogen cycle on oxygen consumption/production or the interactions between oxygen cycle and nitrogen cycle are complex, according to some studies (Sun et al., 2021; Kalvelage et al., 2013; Oschlies et al., 2019). But there is limited information available for the parameterizations of relevant processes. In order to model these processes, we need more field data that allow us to quantify oxygen consumption/production in associated processes and to derive relevant parameters.

Furthermore, there is no sentence on nitrogen reduction processes such as denitrification, whose activity is highly correlated with export production in the ETSP (Kalvelage etal, 2013). As no values are given for NCP, export production, and also the distributions of nutrients or detritus are not provided it is impossible for the reader to judge the quality of the model performance. As far as I know, there has been previously no assessment of their biological component for the depth range below 200 m. In view of the extensive degree of revision, I refrain from specific and technical comments.

**Response**: Thanks for the constructive comments. Because there is limited information available for calibration and validation of some nitrogen reduction processes including denitrification, the regional model does not simulate them at this stage but will include these processes in the future once information becomes available. We have reported/validated many biogeochemical fields, including PP & NCP (Wang *et al.*, 2006b), new production (Wang *et al.*, 2006a) in the euphotic zone, and nitrate, iron, POC/detritus and export production below 200 m (Yu *et al.*, 2021). We have added these references in the revised manuscript.

**Anonymous Referee #2**

Referee comment on "Sensitivity of asymmetric Oxygen Minimum Zones to remineralization rate and mixing intensity in the tropical Pacific using a basin-scale model (OGCM-DMEC V1.2)" by Kai Wang et al., Geosci. Model Dev. Discuss., https://doi.org/10.5194/gmd-2020-431-RC2, 2021

The manuscript, "Sensitivity of asymmetric Oxygen Minimum Zones to remineralization rate and mixing intensity in the tropical Pacific using a basin-scale model (OGCM-DMEC V1.2)" by Wang et al. conduct a suite of model parameter sensitivity experiments with a very old, coarse resolution regional physical ocean model. While using an older model is not necessarily a disadvantage, it is only an advantage is the relative strengths and weaknesses of the model are provided such that the reader can integrate the current analysis to other current understanding. That context is not currently provided. For example, focusing on this region has the advantage that the sponge resets the source O2 (a major weakness of global models) to observations (the authors should note this strength of the current approach). Unfortunately, the comparability of the physical formulation to other models is missing. For example, it is unclear whether the Indonesian Throughflow is represented which is an important part of the advective ventilation in the Western part of the basin and the partitioning of lateral oxygen source waters into the Eastern part of the basin.

**Response**: Thanks for the constructive comments. We have added one sentence about the strengths and weaknesses of the model: "Such model configuration may have a disadvantage for longer simulations and analyses, but has the advantage in reproducing the spatial patterns of most physical and biogeochemical fields". We have also clarified that the model closes the western boundary and no representation of the Indonesian throughflow is included. Consistent with our previous publications (Yu *et al.*, 2021; Wang *et al.*, 2015a; Wang *et al.*, 2008) and numerous other studies (Duteil *et al.*, 2020; Duteil, 2019; Llanillo *et al.*, 2018; Ito and Deutsch, 2013) which focus on the tropical Pacific without the ITF, we rely on the imposed meridional boundary relaxation to constrain our regional solution. Clearly this is inadequate in the strictest sense of the processes mentioned by the reviewer. We posit that the validations presented support our contention that the model is reasonably constrained for the timescales we are considering in this study. Further studies will include the Indonesian-Pacific configuration with an explicit representation of the ITF and DO ventilation into the domain as reported in Rodgers *et al.* (1999). The current focus is on the Pacific processes which we deem are adequately represented in the current configuration.

The analysis uses an inappropriate definition of "suboxic" (see below). Throughout the manuscript the word "rates" is used when "rate constant" is intended (e.g. on line 204 "Reducing remineralization rate by 50% (Cd0.5 minus reference) leads to large decrease...") making it difficult to interpret the result since it is unclear whether the "rate" is proportionally reduced by 50% with fixed concentration or whether there are compensating responses/increases in concentration that result in a change in the remineralization locations. While the result of the combined need to reduce the remineralization rate constant and increase the vertical diffusivity to better match oxygen distributions is encouraging, the manuscript oddly stops there without coming to any implications of the OMZ. What was learned that wasn't known before? Most importantly, the final sentence of the conclusions, "Future studies utilizing advanced models are needed to better understand the impacts of physical and biological interactions on the variability and drivers of the tropical OMZs."

Suggests the authors themselves are unclear as to the significance of the present work to current ocean biogeochemical modeling. As such, I recommend the authors work to clarify there descriptions and the implications and limitations of the current work in revision.

**Response**: Thanks for the constructive comments. Regarding the definition of "suboxic", previous studies have used a wide range of DO as a criterion, e.g., <5 mmol m-3 (Yakushev and Neretin, 1997; Bianchi *et al.*, 2012; Karstensen *et al.*, 2008) and <20 mmol m-3 (Helly and Levin, 2004; Babbin *et al.*, 2015; Wright *et al.*, 2012; Oguz *et al.*, 2000). Some researchers selected DO < 20 mmol m-3 as the boundary of OMZs (Paulmier and Ruiz-Pino, 2009; Fuenzalida *et al.*, 2009; Bettencourt *et al.*, 2015). Accordingly, we adopt the criterion of <20 mmol m-3 for both suboxic water and OMZ volume.

We have made major revisions with an improved approach, i.e., applying a varying constant for OM remineralization that is restricted by oxygen level. We have rewritten the related sections, thus, the questions/comments related 50% are not applicable any more. We have made major revisions with more assessments and in-depth analyses, and rewritten the discussion sections that include the implications and limitations of the current work.

**Technical comments:**

Line 26 – "which made significant progresses" needs rephrasing.

**Response: we have reworded as "with significant progress".**

Line 40 – The authors are misinformed as to the definition of "suboxic", quoting a value of 20 mmol m-3... suboxia is defined as an oxygen level at or below the detection limit, typically 2-10 mmol m-3 where interesting nitrogen redox chemistry such as N2O production, denitrification and annamox occur. The current definition of <20 is rather "strongly hypoxic" as it is well within the detectible range and well above the region of interesting redox chemistry. I would note that the reference the authors cite, Paulmier and Ruiz-Pinu (2009), use a suboxic level of 4.5 umol/gk. Also, if the authors want to describe the truly "suboxic" volume, they should be aware that while Table 3 notes a volume of "suboxic" waters from WOA13, it has been demonstrated that these mapped products strongly underestimate the volume of suboxia at the <5 mmol/m3 definition (Bianchi et al., 2012; https://agupubs.onlinelibrary.wiley.com/doi/full/10.1029/2011GB004209)

**Response**: Thanks for the constructive comments. There is a wide range of DO value used to define suboxic water. In the paper of Paulmier and Ruiz-Pino (2009), they cited DO <4.5 mmol m-3 from an earlier study (Karstensen *et al.*, 2008), but also stated "DO <20 mmol m-3 corresponds to a usual suboxic condition used to separate the aerobic (O2-respiration) from the denitrifying (NO3-respiration) activity (Oguz *et al.*, 2000)". In addition, Wright *et al.* (2012) also defined 0-20 mmol m-3 as suboxic water. We have added some explanation regarding the definition of "suboxic" in the revised manuscript (line 37-39).

We were aware that Fuenzalida *et al.* (2009) and Bianchi *et al.* (2012) used the WOA2005 to estimate OMZ volume, and reported similar values at <20 mmol m-3 definition. In our study, we derived similar OMZ volume (5.97  $10^6$  km3 to the north and 1.43  $10^6$  km3 to the south) using WOA2013.

Line 45-47 – There is an underlying assertion here that data alone provides understanding, and that more availability of data will resolve the underlying mechanisms. This is a false

premise. Only by contextualizing the observations in a theoretical framework can mechanistic understanding be achieved. Also, "our understanding is uncompleted in terms" should be rephrased.

**Response**: We have reworded as "our understanding is limited on the underlying mechanisms that regulate DO dynamics at mid-depth".

Line 54 – "often" seems unnecessary here given that if the OMZ stretches across the equator it would seem to always lead to an overestimate of the OMZ area... unless there is a concomitant decline in area elsewhere in some models. If the latter is indeed the case, it would be worth mentioning. If the intent is just to point out the overestimate, then remove "often".

**Response**: We have removed "often".

Line 57 - "Apparently, it's necessary to..." this is an odd way of saying this, making it sound like the authors are annoyed at the idea.

Response: We have removed "Apparently".

Line 73 – This is a really old, coarse resolution model. A lot of advance has occurred over the last 25 years.

**Response**: While this is an old model, a non-high-resolution model, our previous studies have shown that this model can reproduce mesoscale and sub-mesoscale structures such as the tropical instability wave (TIW) (Zhang *et al.*, 2018; Tian *et al.*, 2018). Indeed, there had been a lot of advance in regional and global model development. We have also made significant improvements in the DMEC sub-model, including the implementation of a phytoplankton dynamic module with a non-steady C:Chl ratio (Wang *et al.*, 2009a), and refined parameterizations of detritus decomposition and DON remineralization (Yu *et al.*, 2021).

Line 74-75 – What is the vertical grid? The stated 10-50m + 20\*10m layers = 210-250 m... this is not deep enough to represent the OMZ...?

**Response**: The vertical resolution varies over depth, with  $\sim$ 30-50 m in the core OMZ (at  $\sim$ 300-500 m) and the total depth of  $\sim$ 1200 m. We have clarified this in the revised manuscript (line 78-81).

Line 75 – What is the longitudinal grid? 150W-80E? Are the walls open to admit the Indonesian throughflow? This would seem critical for representation of O2 ventilation flow into the domain (e.g. Rodgers et al, 1999; https://agupubs.onlinelibrary.wiley.com/doi/abs/10.1029/1998JC900094). Is the Indonesian throughflow prescribed? Both factors should also be explicit.

**Response**: The longitude of our model is from 124°E to 76°W. In this basin-scale model, the western boundary is closed, without representation of the Indonesian throughflow. We have clarified this in the revised manuscript (line 84-85). Consistent with our previous publications and numerous other studies which focus on the tropical Pacific without the ITF, we rely on the imposed meridional boundary relaxation to constrain our regional solution. We

posit that the validations presented support our contention that the model is reasonably constrained for the timescales we are considering in this study.

Line 96 – the parameters here described as "rates" are actually "rate constants", e.g. r is the rate constant for zooplankton respiration.

**Response**: We have made corrections.

Line 116 – What does "DON poor" mean?

Response: We have corrected "DON poor" as "DON pool".

Line 128 – This implies that the model domain extends to 1000 m or more, suggesting line 74-75 is incorrect.

**Response**: Yes, the model domain extends to  $\sim 1200$  m. We have added such information in model description (line 78-79).

Line 140 - It is important to note that "underestimation of supply" is complex and can be from either O2 being too low in the waters that supply or the physical supply mechanisms being either too sluggish or out of balance (e.g. lateral versus vertical and advective versus diffusive"

**Response**: Thank you for the constructive comments. We have made major revisions with more/new analyses on the response of physical supply, including lateral and vertical advection and vertical mixing. In addition, we have also assessed the impacts of biological and physical processes and their interactions/feedbacks on net flux.

Line 145-149 – How did these perturbations influence the fidelity of T and S?

**Response**: Enhanced vertical mixing (i.e., the addition of background diffusion) only applies to the most important variables in this study (i.e., DO and DON). Thus, T and S are not influenced. We have clarified this in the revised manuscript (line 172-173).

Line 152 – What is the reference value of Cd? What does it do? There is no parameter called "Cd" is the appendix, only "CDON0" the remineralization rate constant at 10 C, but it's reference value, 0.001, is very different from 0.5. Looking at Table 1, I see that "Cd05" is actually "CDON0\*0.5". However, it is not clear what the 100-600 m range of "0.0005-0.00025" means... is this the role of temperature on CDON0? This parameter needs a sentence or two of introduction, definition, and contextualization here to avoid confusion.

**Response**: We have made major revisions with an improved approach (i.e., applying a varying value for the constant of OM remineralization). Thus, the questions/comments related Cd0.5 are not relevant any more. We have added "cDON decreases with depth over 100-1000 m, following an exponential function in this study", and also other information in model description section.

Line 204 - the word "rates" is used when "rate constant" is intended (e.g. on line 204 "Reducing remineralization rate by 50% (Cd0.5 minus reference) leads to large decrease...")

making it difficult to interpret the result since it is unclear whether the "rate" is proportionally reduced by 50% with fixed concentration or whether there are compensating responses/increases in concentration that result in a change in the remineralization locations.

**Response**: We have made major revisions with an improved approach to compute the reduced and also varying rates of remineralization. Thus, the above questions/comments are not relevant anymore.

Line 211 – "there is somehow a small decrease..." The use of "somehow" is an insufficient explanation... what is causing this decrease? Is it a response to the remineralization constant decrease?

**Response**: We have made major revisions with new model simulations, new analyses and figures. We have rewritten most paragraphs in Model evaluation and discussions section. Thus, the above questions/comments are not relevant anymore.

Line 224 – Only here is it explained that there was no response in temperature to the diffusivity change. This should have been noted earlier in the results as requested above, as well as the salinity response.

**Response**: Yes, there was no response in temperature. We have clarified this point in the revised manuscript (line 172-173).

Line 228 – "Limited field studies" – why is the defining feature of these studies that they were "limited"? Is the evidence derived from them inconclusive? More explanation of context would be helpful.

**Response**: We have made major revisions, and "Limited field studies" are not in the revised manuscript.

**Anonymous Referee #3**

Referee comment on "Sensitivity of asymmetric Oxygen Minimum Zones to remineralization rate and mixing intensity in the tropical Pacific using a basin-scale model (OGCM-DMEC V1.2)" by Kai Wang et al., Geosci. Model Dev. Discuss., https://doi.org/10.5194/gmd-2020-431-RC3, 2021

This paper examines the sensitivity of the oxygen minimum zones (OMZs) to a change in the remineralization rate and changes in the vertical diffusion coefficient in the tropical Pacific. The goal of this study is to present a calibrated model, to evaluate this model and to identify the mechanisms that explain the asymmetric shape of the OMZ in the tropical Pacific.

Unfortunately this paper only shows the impact of two parameter changes, change in the remineralization rate and the vertical background diffusivity, mainly on the oxygen fields and DON distribution without providing an explanation about the driving mechanisms and a thorough discussion of the results. With this there is so far no scientifically new finding in the current state of the manuscript. In particular an explanation about the mechanisms that drive the asymmetric shape of the OMZs in the tropical Pacific, that is already present in the reference simulation, is missing due to the mainly descriptive nature of the paper. In addition the language that is used is in many places imprecise. E.g. there is no differentiation between vertical and horizontal transport processes as both are termed physical transport.

**Response**: Thank you for the constructive comments. We have made major revisions with more in-depth analyses and thorough discussion of the results, and with explanation about the mechanisms that drive the asymmetry of the OMZs in the tropical Pacific. In particular, we have analyzed the differentiations of physical transport, i.e., horizontal advection, and vertical advection and vertical mixing; we have also analyzed the relative influences of biological and physical processes on sources and sinks of DO, and compared the differences between the northern OMZ and southern OMZ. Our analyses indicate that physical supply has a major influence on the asymmetric OMZs, in which vertical mixing plays a dominant role.

I think there is potential, that this model and the performed sensitivity simulations, could be used to explain the mechanisms that drive the asymmetry in the oxygen fields, although the current version leaves the reader with too many open questions. The model description is not sufficient. There is e.g. no information given about the biogeochemical boundary conditions. What does it mean: All biological components are computed in a manner similar to physical variables. What is the reason to average the model output for the period of 1981-2000 and do not include the last 18 years. Specially, as later in the manuscript some months of year 2009 are compared to some cruise data that are not introduced in the manuscript, they just appear. Are the model data detrended before averaging? Is there a remaining model drift? Why not simply using climatological simulations? What kind of inter-annual forcing is used?

**Response**: Thank you for the constructive comments. We have added more details in the model description section, including biogeochemical boundary conditions (line 81-88), and parameterizations of DON remineralization and detritus decomposition (line 113-120). "All biological components are computed in a manner similar to physical variables" means that all biological variables are influenced by physical processes (e.g., advection and diffusion). We have rephrased that sentence as "All biological components use nitrogen as their unit, in which sources/sinks are determined by biological and chemical processes in

addition to the physical processes (circulation and vertical mixing) that are computed by the OGCM".

We used model output from the period of 1981-2000 for the comparison with WOA2005 dataset because most oxygen data were obtained prior to 2000. Since we did not find much difference between WOA2005 and WOA2013, we did not change the period of 1981-2000 from model simulation in the previous version. But, we use model outputs from 1981-2010 in the revised manuscript. We consider that it is necessary to use cruise data for further model validation, thus we do not using climatological simulations. We have provided more information about the datasets during the revisions.

The model data were not detrended before averaging, and we did not find a model drift. The inter-annual model forcing includes interannual 6-day means of precipitation (ftp://ftp.cdc.noaa.gov/Datasets/gpcp) and surface wind stress (from NCEP reanalysis).

Although it is mentioned at several places (abstract and introduction) that the model is calibrated - there is no further information given how. Is it calibrated only against the oxygen fields? As the reference simulation already represents the asymmetric shape of OMZ, I assume that this is at least partly a result of the model calibration. The only information that is in the paper: Some parameters have been changed compared to an earlier version of the model. This is however not sufficient.

**Response**: Thank you for the constructive comments. The basin-scale model was calibrated against many biogeochemical fields, including chlorophyll (Wang *et al.*, 2009a), nitrogen cycle (Wang *et al.*, 2009c) and carbon cycle (Wang *et al.*, 2015a). In this study, we further calibrated against DON remineralization and oxygen fields. Although the reference simulation represents the asymmetric OMZ, it is not a result of the model calibration against biogeochemical fields. Our analyses indicate that it is largely determined by physical processes. We have added more information regarding model calibration, and in-depth analyses of physical transport during the revision. We have also provided more details on model description, including parameterizations of relevant biogeochemical processes (line 102-120).

The model evaluation and validation is lacking. The oxygen fields look fine, but there is no information about, e.g. the circulation. How good is the current system represented in this model? What about nitrogen and phosphate - is there a bias in the east west gradient (one of the problems as shown e.g. by Dietze and Loeptien, 2013, doi:10.1002/gbc.20029) or is that reduced etc.

**Response:** Our previous studies have provided model validation for chlorophyll (Wang *et al.*, 2009a; Wang *et al.*, 2013a), nitrogen cycling (Wang *et al.*, 2009b), and carbon fields (Wang *et al.*, 2015a; Yu *et al.*, 2021). The model does a good job in representing the current system thus physical fields such as sea surface temperature (Wang *et al.*, 2008; Zhang *et al.*, 2018) (also see Figure a & c below). The model can also reproduce the west-to-east gradient of nitrate along the equator (see Figure b & d below). We have made a lot of improvement in model evaluation during the major revisions, including (1) the responses of biological consumption and physical supply to reduced O:C utilization ratio and enhanced vertical mixing, (2) interactive effects of physical and biological processes on asymmetric OMZs.

Regarding the structure of the paper - I would suggest to reorganize the paper: There should be a clear separation between the model set up as well as the set up of the sensitivity simulations and the results. In addition, regarding the comparison with the cruise data - these simply appear, where do they come from? Same happens with the DON data from HOT.

**Response**: Thank you for your suggestion. We have made major revisions with some reorganization by taking into account all reviewers' comments and suggestions. We have made changes in the Model experiments and validation section, and added the source of cruise data (line 201-202).

The sensitivity simulations show an improved representation of the OMZ in Fig4. The description of this improvement in the text is somewhat incorrect. The asymmetric shape is present in all simulations. It seems that the changes in the parameters result in an overall increase of DO and not necessarily an increase of the asymmetry. As at the southern hemisphere the DO concentrations are lower (in fact higher), one might get the impression that this increase might be slightly larger, but there is no evidence that this is the case at this stage. In addition, in the text it is stated that between 2°S-2°N the values of DO are relatively high (~30-40 mmol m-3) in Cd0.5Kb0.5. Fig 4f does not support this. Around 400m depth the DO concentrations are below 20 mmol m-3.

**Response**: Thank you for the constructive comments. We realize the incorrect text (i.e.,  $\sim$ 30-40 mmol m-3), and have made corrections. We have made major revisions, with in-depth analyses and discussion of changes/responses of sources and sinks between the south and north OMZs. Our analyses demonstrate that net flux (i.e., increase of DO) is greater to the south than to the north, and the asymmetry of OMZs is largely determined by physical supply, in which vertical mixing plays a dominant role. We have rewritten the relevant sections in the revised manuscript.

Fig 10 clearly shows differences - but there is no explanation given about the choice of the region shown as well as what have been done with the data - I guess they have been averaged. As the vertical oxygen gradients are different in these two regions, I would have expected a difference in the oxygen response, as the vertical diffusive transport depends on the gradient. Unfortunately the explanation of the results are again left to the reader.

**Response**: Thank you for the constructive comments. We have made major revisions, with new analyses (and figures) showing the difference in DO response, and changes in consumption, total supply and horizontal and vertical advections and vertical mixing (Figures 9-11). We have rewritten the relevant section with thorough discussion and explanation on the differences in biological consumption and physical supply between the southern and northern OMZs.

In addition there is no clear explanation given about the choice of the parameter change for the sensitivity simulations and with this it seems rather arbitrary. Also why is an increase of the vertical background diffusivity of  $0.5 \text{ cm}^2/\text{s}$  optimal, what about higher rates?

**Response**: Thank you for the constructive comments. We have added explanation about the choice of the parameter changes in the revised manuscript. New biological parameterizations were derived based on field measurement (line 163-166). Regarding the parameter for background diffusion, there is a large range (~0.01-0.5 cm2/s) in other modeling studies. Most studies reported improvements in mid-depth DO, using a value <0.5 cm2/s, e.g., Duteil and Oschlies (2011). While stronger vertical mixing could also lead to improved oxygen at mid-depth, a too-high value for background diffusion (mainly representing molecular diffusion) may not be acceptable.

A thorough discussion of the results is missing. How are the results related to other modeling studies? The sensitivity studies show that the major changes occur along the equator - so this indicates that somehow the representation of the current system is important and needs to be shown and discussed. There are several physical processes in addition to the known impact of vertical mixing that seem to be capable to reduce the oxygen model bias along the equator, e.g. enhancing zonal diffusion or enhancing viscosity.

**Response**: Thank you for the constructive comments. We have made major revisions with more in-depth analyses and thorough discussion of the results. We have included comparisons with other modeling studies, e.g., zonal advection and meridional advection, vertical advection and vertical mixing (line 280-283). We have also provided more information on model validation, including physical fields (e.g., a good representation of the current system). We agree that other physical processes in addition to vertical mixing may be able to reduce the model bias along the equator. But, this is beyond the scope of this study. Future studies will include sensitivity experiments with enhanced zonal diffusion or viscosity.

Also, as the model is forced with an inter-annually varying forcing, what about the potential impact of El Nino events?

**Response**: Our preliminary analyses indicated that there were no significant impacts of El Nino events on mid-depth oxygen. But, El Nino events might have large impacts on oxygen fields in surface water. These analyses will be carried out in a future study.

As the manuscript needs substantial revisions, I do not add any specific and/or technical comments at this stage of the review process.

**Reference**

Babbin, A. R., Bianchi, D., Jayakumar, A. & Ward, B. B. (2015). Rapid nitrous oxide cycling in the suboxic ocean. *Science* 348: 1127-1129.

- Bettencourt, J. H., Lopez, C., Hernandez-Garcia, E., Montes, I., Sudre, J., Dewitte, B., Paulmier, A. & Garcon, V. (2015). Boundaries of the Peruvian oxygen minimum zone shaped by coherent mesoscale dynamics. *Nature Geoscience* 8(12): 937-U967.
- Bianchi, D., Dunne, J. P., Sarmiento, J. L. & Galbraith, E. D. (2012). Data-based estimates of suboxia, denitrification, and N2O production in the ocean and their sensitivities to dissolved O2. *Global Biogeochemical Cycles* 26(2): 1-13.
- Duteil, O. (2019). Wind Synoptic Activity Increases Oxygen Levels in the Tropical Pacific Ocean. *Geophysical Research Letters* 46(5): 2715-2725.
- Duteil, O., Frenger, I. & Getzlaff, J. (2020). Intermediate water masses, a major supplier of oxygen for the eastern tropical Pacific ocean. *Ocean Science*.
- Duteil, O. & Oschlies, A. (2011). Sensitivity of simulated extent and future evolution of marine suboxia to mixing intensity. *Geophysical Research Letters* 38(6).
- Fuenzalida, R., Schneider, W., Garces-Vargas, J., Bravo, L. & Lange, C. (2009). Vertical and horizontal extension of the oxygen minimum zone in the eastern South Pacific Ocean. *Deep-Sea Research Part Ii-Topical Studies in Oceanography* 56(16): 1027-1038.
- Helly, J. J. & Levin, L. A. (2004). Global distribution of naturally occurring marine hypoxia on continental margins. *Deep-Sea Research Part I-Oceanographic Research Papers* 51(9): 1159-1168.
- Ito, T. & Deutsch, C. (2013). Variability of the oxygen minimum zone in the tropical North Pacific during the late twentieth century. *Global Biogeochemical Cycles* 27(4): 1119-1128.
- Kalvelage, T., Lavik, G., Lam, P., Contreras, S., Arteaga, L., Löscher, C. R., Oschlies, A., Paulmier, A., Stramma, L. & Kuypers, M. M. M. (2013). Nitrogen cycling driven by organic matter export in the South Pacific oxygen minimum zone. *Nature Geoscience* 6(3): 228-234.
- Karstensen, J., Stramma, L. & Visbeck, M. (2008). Oxygen minimum zones in the eastern tropical Atlantic and Pacific oceans. *Progress in Oceanography* 77(4): 331-350.
- Llanillo, P. J., Pelegri, J. L., Talley, L. D., Pena-Izquierdo, J. & Cordero, R. R. (2018). Oxygen Pathways and Budget for the Eastern South Pacific Oxygen Minimum Zone. *Journal of Geophysical Research: Oceans* 123(3): 1722-1744.
- Oguz, T., Ducklow, H. W. & Malanotte-Rizzoli, P. (2000). Modeling distinct vertical biogeochemical structure of the Black Sea: Dynamical coupling of the oxic, suboxic, and anoxic layers. *Global Biogeochemical Cycles* 14(4): 1331-1352.
- Oschlies, A., Koeve, W., Landolfi, A. & Kahler, P. (2019). Loss of fixed nitrogen causes net oxygen gain in a warmer future ocean. *Nat Commun* 10(1): 2805.

- Paulmier, A. & Ruiz-Pino, D. (2009). Oxygen minimum zones (OMZs) in the modern ocean. *Progress in Oceanography* 80(3-4): 113-128.
- Rodgers, K. B., Cane, M. A., Naik, N. H. & Schrag, D. P. (1999). The role of the Indonesian Throughflow in equatorial Pacific thermocline ventilation. *Journal of Geophysical Research: Oceans* 104(C9): 20551-20570.
- Sun, X., Frey, C., Garcia-Robledo, E., Jayakumar, A. & Ward, B. B. (2021). Microbial niche differentiation explains nitrite oxidation in marine oxygen minimum zones. *ISME J* 15(5): 1317-1329.
- Tian, F., Zhang, R. H. & Wang, X. (2018). A Coupled Ocean Physics Biology Modeling Study on Tropical Instability Wave - Induced Chlorophyll Impacts in the Pacific. *Journal of Geophysical Research: Oceans* 123(8): 5160-5179.
- Wang, X. J., Behrenfeld, M., Le Borgne, R., Murtugudde, R. & Boss, E. (2009a). Regulation of phytoplankton carbon to chlorophyll ratio by light, nutrients and temperature in the Equatorial Pacific Ocean: a basin-scale model. *Biogeosciences* 6(3): 391-404.
- Wang, X. J., Christian, J. R., Murtugudde, R. & Busalacchi, A. J. (2006a). Spatial and temporal variability in new production in the equatorial Pacific during 1980-2003: Physical and biogeochemical controls. *Deep-Sea Research Part II* 53(5-7): 677-697.
- Wang, X. J., Christian, J. R., Murtugudde, R. & Busalacchi, A. J. (2006b). Spatial and temporal variability of the surface water pCO(2) and air-sea CO2 flux in the equatorial Pacific during 1980-2003: a basin-scale carbon cycle model. *Journal of Geophysical Research* 111(C7): C07S04, doi:10.1029/2005JC002972.
- Wang, X. J., Le Borgne, R. & Murtugudde, R. (2009b). Nitrogen uptake and regeneration pathways in the equatorial Pacific: a basin scale modeling study. *Biogeosciences* 6: 2647-2660.
- Wang, X. J., Le Borgne, R., Murtugudde, R., Busalacchi, A. J. & Behrenfeld, M. (2008). Spatial and temporal variations in dissolved and particulate organic nitrogen in the equatorial Pacific: biological and physical influences. *Biogeosciences* 5(6): 1705-1721.
- Wang, X. J., Murtugudde, R., Hackert, E. & Maranon, E. (2013a). Phytoplankton carbon and chlorophyll distributions in the equatorial Pacific and Atlantic: A basin-scale comparative study. *Journal of Marine Systems* 109: 138-148.
- Wang, X. J., Murtugudde, R., Hackert, E., Wang, J. & Beauchamp, J. (2015a). Seasonal to decadal variations of sea surface pCO2 and sea-air CO2 flux in the equatorial oceans over 1984–2013: A basin-scale comparison of the Pacific and Atlantic Oceans. *Global Biogeochemical Cycles* 29: 597-609.

- Wang, X. J., Murtugudde, R., Hackert, E., Wang, J. & Beauchamp, J. (2015b). Seasonal to decadal variations of sea surface pCO(2) and sea-air CO2 flux in the equatorial oceans over 1984-2013: A basin-scale comparison of the Pacific and Atlantic Oceans. *Global Biogeochemical Cycles* 29(5): 597-609.
- Wang, X. J., Murtugudde, R. & Le Borgne, R. (2009c). Nitrogen uptake and regeneration pathways in the equatorial Pacific: a basin scale modeling study. *Biogeosciences* 6(11): 2647-2660.
- Wang, X. J., MurtuguddeA, R., Hackert, E. & Maranon, E. (2013b). Phytoplankton carbon and chlorophyll distributions in the equatorial Pacific and Atlantic: A basin-scale comparative study. *Journal of Marine Systems* 109: 138-148.
- Wright, J. J., Konwar, K. M. & Hallam, S. J. (2012). Microbial ecology of expanding oxygen minimum zones. *Nature Reviews Microbiology* 10(6): 381-394.
- Yakushev, E. V. & Neretin, L. N. (1997). One-dimensional modeling of nitrogen and sulfur cycles in the aphotic zones of the Black and Arabian Seas. *Global Biogeochemical Cycles* 11(3): 401-414.
- Yu, J., Wang, X., Murtugudde, R., Tian, F. & Zhang, R. H. (2021). Interannual to Decadal Variations of Particulate Organic Carbon and the Contribution of Phytoplankton in the Tropical Pacific During 1981 2016: A Model Study. *Journal of Geophysical Research: Oceans* 126(1).
- Zhang, R. H., Tian, F. & Wang, X. J. (2018). A New Hybrid Coupled Model of Atmosphere, Ocean Physics, and Ocean Biogeochemistry to Represent Biogeophysical Feedback Effects in the Tropical Pacific. *Journal of Advances in Modeling Earth Systems* 10(8): 1901-1923.

---

## Referee Report (RR1)

Review:

**Sensitivity of asymmertric Oxygen Minimum Zones to mixing intensity and stoichiometry in the tropical Pacific using a basin-scale model (OGCM-DMEC V1.4)  by K. Wang et al.**

submitted to GMD

This is the second review of the paper and the manuscript has improved significantly compared to the initial submission. However there are some remaining comments that need to be taken into account, before it can be considered for publication. The differentiation between physical, biogeochemical and coupled feedback mechanisms is still not shown in a sufficient way. The experimental setup allows to show the inter-dependecies between physical and biogeochemical processes in more detail than it is done. I would further rate this to be the main scientific impact of this study rather than allocating that to future studies. The general conclusion - physics have a big impact and there are complicated interactions - are too general and doesn't bring new insight into the topic. In particular processes in the depth range 400-700m, that are of key importance for the OMZ, are explained only in a perfunctorily manner. I would guess that a great deal can be done by changing Fig 10-12 and show here the same difference of simulations as done e.g. in Fig 9. Then you are able to differentiate between a change in the physical supply that results from a biogeochemical process and value/rate that against the change of the physical process and the combined change. The limitation of the study is clear, that only two processes are considered here, but the impact of these two processes should be fully investigated and explained here.

Further comments:

1. Abstract line 17 - this result is logically inconsistent: DO is more sensitive to biological processes between 200-700m and to physical processes between 400-1000m $\rightarrow$ so what is the case in the region between 400-700m which is the key region where models show large representation deficiencies?

2. ll 26-28 - cite missing - as the carbon cycle has raise much attention ...

3. ll 68-69: A motivation why these two processes are chosen is missing. In particular it should be mentioned that vertical mixing is not added to the model as a physical process in general, but only to two of the tracers, as well as the resulting consequences of this choice.

4. l 149: I guess this is a typo and the reference run is not model version V1.2, but V1.4

5. I have a problem with Fig 3 - the fittet functions seems a bit random ... I like the idea of taking two different sites to get an estimate, but the fit itself is not convincing, in particular for Fig. 3b. Is there some kind of weighting applied? What fitting method has been used? In addition, how does the kinetic function that is derived and added to equation 5 look like - what is the finally used equation for the introduced sensitivity simulations?

6. ll 172-173: I am struggeling with the approach that addtional diffusion is not added to all tracers in the model, but solely to DO and DON. This approach needs a justification. What are the potential consequences when adding the background diffusion only to DO and DON? How does it look like when additional background diffusion is added to all tracers in the model? In addition it is important to clarify this in the abstract as well as in the conclusions.

7. Fig 4: Is that averaged between 120°W and 90°W? Please specify.

8. From Fig 5 I cannot deduce that Km18.8Kb0.5 gives the best results as it is done in the text. I would agree for the correlation, but by eye it seems that this is not the case for the standard deviation. This needs to be specified/clarified in the text.

9. ll 193 - 199: Why did you choose the threshold 20 and 60 mmol m-3? You choose the same range for the added background diffusion as Duteil and Oschlies (2011) used, and they found a tipping point, where the suboxic volume starts decreasing when further increasing the diffusivity, but for a much lower threshold. Is that the same case for your model? What does that mean regarding your best choice?

10. Fig 6: Figure caption needs to explain the figure without referencing to the text.

11. ll 229-230: Same sentence as in the Abstract. Please be more specific for the depth rage 400-700m. It's the key region of interest and currently it is treated not as such. Are both processes of equal importance, is one more important than the other or is this all just a result of how the background diffusion is applied?

12. Fig 9 shows differences in the northern and southern hemispheric responses in mid depth - why do you find this relatively large impact of the reduced O:C utilization ratio in the region of the northern OMZ?

13. Fig 10 caption - what is shown? An average between 120°W and 90°W? Black contour lines? Please be more precise. And I don't understand what you intend to show by the difference plots such as d-a, that is Km18.7kb0.5-Km18-Kb0.5+Ref- ... if the responses would be linear, then the result would be Ref ...

14. How is the physical supply estimated? I couldn't find the relevant information in the manuscript.

15. How is the biological consumption estimated?

16. ll 264-165: what are the various physical, biological and chemical processes? Please explain the potential processes that lead to the results shown in Fig 10.

17. l 281 - there is no Fig S2 - I guess you mean S1 ...

18. I would suggest that you refer all your difference plots to the reference run. So when you would like to show the impact of reduced O:C utilisation then you show km18.7-Ref (as you do), when you would like to show the impact of added vertical mixing (to DO and DON) then you show kb0.5-Ref and when you want to show the combined impact it's Km18.7kb0.5-Ref. That's how you started when showing changes in the oxygen concentration and it would make it much easier for the reader to follow.

19. l 284: How does Ref look like in this case - these difference of the difference plots should be close to Ref.

20. l 284-285: Fig 12 e and f: The small difference between these plots just shows that the applied changed add up almost linearly ... KM18.7-Ref is supposed to show the impact of the reduced O:C utilisation and the same is done by Km18.7Kb0.5 - Kb0.5

21. l 285-286: Why is this the case? Mixing depends on the gradient - how does the reduced O:C utilisation change that?

22. ll 289-290: I am not convinced that 12j is showing that - How does mixing look like in Ref?

23. l 303 and l 315 (probably also in other places): The use of "biological parameter" is incorrect, the parameters that are used in the model are, so far as I see spatially and temporally constant values, wheres the resulting tracer distributions show differences.

24. I appreciate the added sections 4.4 and 4.5. However the discussion is in the current version not very clear and easy to read and needs to be rewritten: Is this only taking the reference version into account? From which simulation are the DON concentrations taken that are mentioned here? Implication of current research should be the differentiation between the biogeochemical and physical impact in mid-depth (400-700m). Limitation of the study is that two explicit processes have been investigated. These should be set into relation to other processes of potential importance.

---

## Author Response (AR2)

**Responses to the reviewers' comments:**

**Reviewer 1**

I am generally satisfied with the revision. My only concern is the paragraph introductory sentence on lines 315-316, "It appears that the asymmetric distributions differ largely between biological parameters, and there are almost opposite patterns between oxygen consumption (or DOM remineralization) and DOM concentration". I do not understand what the authors are trying to convey. Are the authors trying to say that observations suggest that the model cannot simultaneously simulate the concentrations with the same set of biological parameters? From what model or observational constraint does the phrase "It appears" come from?

**Response**: Thank you for the constructive comments. It should be "biological fields" not "biological parameters". We have revised as "It appears that the asymmetric distributions differ largely between biological fields in the tropical Pacific. In particular, there are almost opposite patterns between oxygen consumption (or DOM remineralization) and DOM concentration, which may be attributed to the difference in the rate of DOM remineralization between north and south". This sentence is not only an introductory sentence for the second paragraph but also a summary of the first paragraph.

**Reviewer 2**

This is the second review of the paper and the manuscript has improved significantly compared to the initial submission. However there are some remaining comments that need to be taken into account, before it can be considered for publication. The differentiation between physical, biogeochemical and coupled feedback mechanisms is still not shown in a sufficient way. The experimental setup allows to show the inter-dependecies between physical and biogeochemical processes in more detail than it is done. I would further rate this to be the main scientific impact of this study rather than allocating that to future studies. The general conclusion - physics have a big impact and there are complicated interactions - are too general and doesn't bring new insight into the topic. In particular processes in the depth range 400-700m, that are of key importance for the OMZ, are explained only in a perfunctorily manner. I would guess that a great deal can be done by changing Fig 10-12 and show here the same difference of simulations as done e.g. in Fig 9. Then you are able to differentiate between a change in the physical supply that results from a biogeochemical process and value/rate that against the change of the physical process and the combined change. The limitation of the study is clear, that only two processes are considered here, but the impact of these two processes should be fully investigated and explained here.

**Response**: Thank you for the constructive comments. We have made major revisions to address all the comments/suggestions. In particular, we have conducted further analyses (with new/revised figures, as suggested) and discussions, with some rewriting on the differentiation between physical, biogeochemical and coupled feedback mechanisms (section 4.4). We have carefully re-evaluated the analyses on the relative roles of physical and biological processes, and made corrections regarding the depth range 400-700 m (see further explanation/responses below).

**Further comments:**

1. Abstract line 17 - this result is logically inconsistent: DO is more sensitive to biological processes between 200-700 m and to physical processes between 400-1000 m  $\rightarrow$  so what is

the case in the region between 400-700 m which is the key region where models show large representation deficiencies?

**Response:** Thank you for the constructive comments. Our previous statement "DO is more sensitive to biological processes between 200-700 m and to physical processes between 400-1000 m" is not really correct or accurate. We have corrected as "DO is more sensitive to biological processes between 200-400 m but to physical processes below 400 m".

2. ll 26-28 - cite missing - as the carbon cycle has raise much attention ...

Response: We have added some references related to carbon cycle (line 27-28).

3. Il 68-69: A motivation why these two processes are chosen is missing. In particular it should be mentioned that vertical mixing is not added to the model as a physical process in general, but only to two of the tracers, as well as the resulting consequences of this choice.

**Response:** Thank you for the constructive comments. We have revised the introduction to emphasize the motivation (line 59-67). We have also mentioned "The reference run applied a zero value for background diffusion (see eq. 9). However, a previous modelling study demonstrated that vertical background diffusion was an important process for DO supply at mid-depth (Duteil and Oschlies, 2011). Accordingly, we conduct a sensitivity experiment to test a set of values for background diffusion (Kb as 0.1, 0.25 and 0.5 cm2 s-1). The addition of background diffusion is only applied to the two key variables (DO and DON) in this analysis to eliminate any potential interactions and feedbacks between various physical and biogeochemical processes (note: our model experiments showed no significant effects on modelled DO dynamics with background diffusion applied to the nutrients)".

**4. 1 149: I guess this is a typo and the reference run is not model version V1.2, but V1.4 **Response:** We have corrected as V1.4.**

5. I have a problem with Fig 3 - the fittet functions seems a bit random ... I like the idea of taking two different sites to get an estimate, but the fit itself is not convincing, in particular for Fig. 3b. Is there some kind of weighting applied? What fitting method has been used? In addition, how does the kinetic function that is derived and added to equation 5 look like - what is the finally used equation for the introduced sensitivity simulations?

**Response:** We used the fitting method of least squares, and did not apply any kind of weighting. But, for the old Fig. 3b, we only used four data points (excluding the smallest value) because the fitting curve using all five data points is too far away from the most data points (see the black line in the new Fig. 3b). It seems that the curve with Km=6.9 fits well for both sites, but our model sensitivity experiments show that applying Km=18.7 gets better performance for DO fields.

6. Il 172-173: I am struggeling with the approach that additional diffusion is not added to all tracers in the model, but solely to DO and DON. This approach needs a justification. What are the potential consequences when adding the background diffusion only to DO and DON? How does it look like when additional background diffusion is added to all tracers in the model? In addition it is important to clarify this in the abstract as well as in the conclusions.

**Response:** Thank you for the constructive comments. Our model experiments showed no significant effects on modelled DO distribution (see figure below) with background diffusion applied to dissolved iron (Fe) and nitrate (Ni). We have added "The addition of background diffusion is only applied to the two key variables (DO and DON) in this analysis to eliminate any potential interactions and feedbacks between various physical and biogeochemical processes (note: our model experiments showed no significant effects on modelled DO dynamics with background diffusion applied to the nutrients)" in the text.

---

## Author Response (AR3)

Wang et al. present a basin-scale ocean circulation model coupled to a dynamic ecosystem carbon cycle model which they apply to the tropical Pacific in order to assess the role of the parameterizations of vertical mixing and of the oxygen uptake during DOM remineralisation for the asymmetric shape of the OMZs in the tropical Pacific.

I understand that this manuscript has already gone through several review rounds. I have gone through the replies to the comments made at the previous round and find that the authors have satisfactorily replied to the comments of the two reviewers. The text has been extensively rewritten.

There are only a few minor questions and comments that I would like to make at this stage.

Line 155: why "Monthly"?
Response: We have removed "monthly".

Lines 208-209: "700-100 m" should be "700-1000 m", I guess.
Response: We have corrected as "700-1000 m".

I have one question about the nutrient consumption rate terms in equations (B8) and (B9), not raised in the previous reviews, as far as I can see. At first both look like classical Monod terms. However, they are more complicated than that, as the $N\_S\_UP$ and $N\_L\_UP$ are not concentrations and $A\_UP$ is not a half-saturation constant: $N\_S\_UP$ and $N\_L\_UP$ (from equations (B11) and (B12), resp.) are actually rate-law expressions themselves, composed of a hyperbolic function (Monod factor) in $NO\_3$ (with half-saturation constants $K\_S\_NO3$ and $K\_L\_NO3$, resp.) multiplied by a hyberbolic inhibition factor depending on $NH\_4$, with an inhibition contant $K\_NH\_4$); $A\_UP$ is a Monod factor in $NH\_4$, which uses $K\_NH\_4$ as a half-saturation constant. I have never seen this nested usage of Monod factors and Monod factors with inhibition in Monod functions before. The resulting uptake rate expressions look highly non-linear to me and I expect them to come close to their maximum values only in very narrow concentration ranges and leading to small values for most $NO\_3$-$NH\_4$ concentration combinations. What is the rationale behind these rather convoluted formulations?
Response: Thank you for the constructive comments. Equations (B8-B9, B11-B13) are for the uptakes of ammonium and nitrate, with a preferential uptake of ammonium, which are the same as those (eq. 8 and eq. 9) used in Vallina and Le Quéré (2008).
We used Monod functions for the growth rates of small and large phytoplankton (see eq. B14- B17).

$$Q_A = \frac{A}{k_A + A} \qquad\qquad (8)$$

$$Q_N = \frac{N \cdot (1 - (A/k_A + A))}{k_N + N} \qquad\qquad (9)$$

There are a few typesetting problems with symbols used in the equations (B3) and following and their names in the table named Appendix B.

Response: We have corrected the typesetting problems with symbols used in the equations (B1-B4, B9) and other places.

Please notice also the repeated "zootoplankton" instead of "zooplankton" in that table.

Response: We have corrected as "zooplankton".

Finally: please have the text proof-read by a proficient English speaker/writer. There remain numerous English errors (missing articles, ill-formed sentences, ...).

Response: We have carefully checked the whole manuscript and had a proof-read by a proficient English speaker/writer.

Reference:

Vallina, S. M. and Le Quéré, C.: Preferential uptake of over in marine ecosystem models: A simple and more consistent parameterization, Ecological Modelling, 218, 393-397, 2008.